# Quantifying the impact of quarantine duration on COVID-19 transmission

**Peter Ashcroft[1]\*, Sonja Lehtinen[1], Daniel C Angst[1], Nicola Low[2], Sebastian Bonhoeffer[1]\***

[1]Institute of Integrative Biology, ETH Zürich, Zürich, Switzerland; [2]Institute of Social and Preventive Medicine, University of Bern, Bern, Switzerland

**Abstract** The large number of individuals placed into quarantine because of possible severe acute respiratory syndrome coronavirus 2 (SARS CoV-2) exposure has high societal and economic costs. There is ongoing debate about the appropriate duration of quarantine, particularly since the fraction of individuals who eventually test positive is perceived as being low. We use empirically determined distributions of incubation period, infectivity, and generation time to quantify how the duration of quarantine affects onward transmission from traced contacts of confirmed SARS-CoV-2 cases and from returning travellers. We also consider the roles of testing followed by release if negative (test-and-release), reinforced hygiene, adherence, and symptoms in calculating quarantine efficacy. We show that there are quarantine strategies based on a test-and-release protocol that, from an epidemiological viewpoint, perform almost as well as a 10-day quarantine, but with fewer person-days spent in quarantine. The findings apply to both travellers and contacts, but the specifics depend on the context.

## Introduction

Quarantining individuals with high risk of recent infection is one of the pillars of the non-pharmaceutical interventions to control the ongoing severe acute respiratory syndrome coronavirus 2 (SARS-CoV-2) pandemic (*Kucharski et al., 2020*). Owing to the large fraction of transmission of SARS-CoV-2 that is pre-symptomatic or asymptomatic (*Ashcroft et al., 2020*; *Buitrago-Garcia et al., 2020*; *Ferretti et al., 2020b*; *He et al., 2020*), quarantine can prevent a substantial fraction of onward transmission that would not be detected otherwise. In mathematical modelling studies, it was estimated that thermal screening at airports would allow more than 50% of infected travellers to enter the general population (*Quilty et al., 2020*; *Gostic et al., 2020*), which could have been prevented by mandatory quarantine. Quarantine is also a fundamental part of the test–trace–isolate–quarantine (TTIQ) intervention strategy to break chains of transmission within a community (*Salathé et al., 2020*). With the high or increasing case numbers that are observed in many places around the globe, however, more and more people are being placed into quarantine.

There is ongoing public debate about the appropriateness of quarantine and its duration. Quarantine lowers onward transmission in two ways: first, preventing transmission prior to symptom onset (with the assumption that symptomatic individuals will isolate) and decreasing overall transmission from persistently asymptomatic individuals. The appropriate length of quarantine thus depends on both incubation period and the temporal profile of infectiousness. In theory, quarantine periods could be avoided altogether through widespread and regular testing programmes, but the low sensitivity of reverse transcriptase PCR (RT-PCR) tests, particularly in early infection (*Kucirka et al., 2020*), as well as limitations on testing capacity in most countries preclude this approach. Quarantine has high economic, societal, and psychological costs (*Nicola et al., 2020*; *Brooks et al., 2020*). It restricts individual freedoms (*Parmet and Sinha, 2020*), although the level of restriction imposed is generally judged to be proportionate, given the severity of coronavirus disease 2019 (COVID-19).

**\*For correspondence:**
peter.ashcroft@env.ethz.ch (PA);
seb@env.ethz.ch (SB)

**Competing interests:** The authors declare that no competing interests exist.

**eLife digest** The COVID-19 pandemic has led many countries to impose quarantines, ensuring that people who may have been exposed to the SARS-CoV-2 virus or who return from abroad are isolated for a specific period to prevent the spread of the disease. These measures have crippled travel, taken a large economic toll, and affected the wellbeing of those needing to self-isolate. However, there is no consensus on how long COVID-19 quarantines should be.

Reducing the duration of quarantines could significantly decrease the costs of COVID-19 to the overall economy and to individuals, so Ashcroft et al. decided to examine how shorter isolation periods and test-and-release schemes affected transmission. Existing data on how SARS-CoV-2 behaves in a population were used to generate a model that would predict how changing quarantine length impacts transmission for both travellers and people who may have been exposed to the virus. The analysis predicted that shortening quarantines from ten to seven days would result in almost no increased risk of transmission, if paired with PCR testing on day five of isolation (with people testing positive being confined for longer). The quarantine could be cut further to six days if rapid antigen tests were used.

Ashcroft et al.'s findings suggest that it may be possible to shorten COVID-19 quarantines if good testing approaches are implemented, leading to better economic, social and individual outcomes.

The low number of individuals placed in quarantine that turn out to be infected is another argument that is given against quarantine.

Individuals are generally placed into quarantine for one of two reasons: either they have been identified as a recent close contact of a confirmed SARS-CoV-2 case by contact tracing, or they have returned from recent travel to an area with community transmission that has been assessed to pose significant epidemiological risk (**WHO, 2020**). These groups of quarantined individuals differ in two important ways: compared with traced contacts, travel returners may have lower probability of being infected and have less precise information about the likely time of exposure. This raises the question whether the duration of quarantine should be the same for these two groups of individuals.

To our knowledge, there are no published analyses of surveillance data that directly assess the impact of duration of quarantine on SARS-CoV-2 transmission (**Nussbaumer-Streit et al., 2020**). In this study, we present a mathematical model that allows quantification of the effects of changing quarantine duration. We use the distributions of incubation time (time from infection to onset of symptoms), infectivity (infectiousness as a function of days since symptom onset), and generation time (difference of timepoints of infection between infector and infectee). These distributions have been estimated by **Ferretti et al., 2020b**, combining multiple empirical studies of documented SARS-CoV-2 transmission pairs (**Ferretti et al., 2020a**; **Xia et al., 2020**; **Cheng et al., 2020**; **He et al., 2020**).

Using the model, we explore the effect of duration of quarantine for both traced contacts of confirmed SARS-CoV-2 cases and for returning travellers on the fraction of prevented onward transmission. We assess the effects of test-and-release strategies and the time delay between test and result. These considerations are particularly important given that multiple testing has been shown to be of little benefit (**Clifford, 2020**). We also address the role of pre-symptomatic patients becoming symptomatic and therefore being isolated independent of quarantine. Furthermore, as one of the arguments for shortening the duration of quarantine is to increase the number of people complying with the recommendation, we investigate by how much adherence needs to increase to offset the increased transmission due to earlier release from quarantine. Finally, we assess the role of reinforced individual-level prevention measures, such as mask wearing, for those released early from quarantine.

Making policy decisions about the duration of quarantine fundamentally requires specifying how the effectiveness of quarantine relates to its costs. The effectiveness can be measured in terms of the overall reduction of transmission, while economic, societal, and individual costs are likely a function of the number of days spent in quarantine. In addition to the epidemiological outcome, which

considers only the reduction in transmission, we also present results based on the ratio of transmission prevented to the average number of days spent in quarantine.

## Results

### Model description

In the absence of quarantine, individuals that are infected with SARS-CoV-2 can infect further individuals in the population. In the model, the timing of onward transmission from an infected individual is determined by the generation time distribution, which describes the time interval between the infection of an infector and infectee (see *Figure 1—figure supplement 1*). To quantify how much transmission is prevented by quarantining individuals who have been infected with SARS-CoV-2, we need to know the time at which the individual was exposed ($t_E$), as well as when they enter ($t_Q$) and are released from ($t_R$) quarantine. The fraction of transmission that is prevented by quarantine is then the

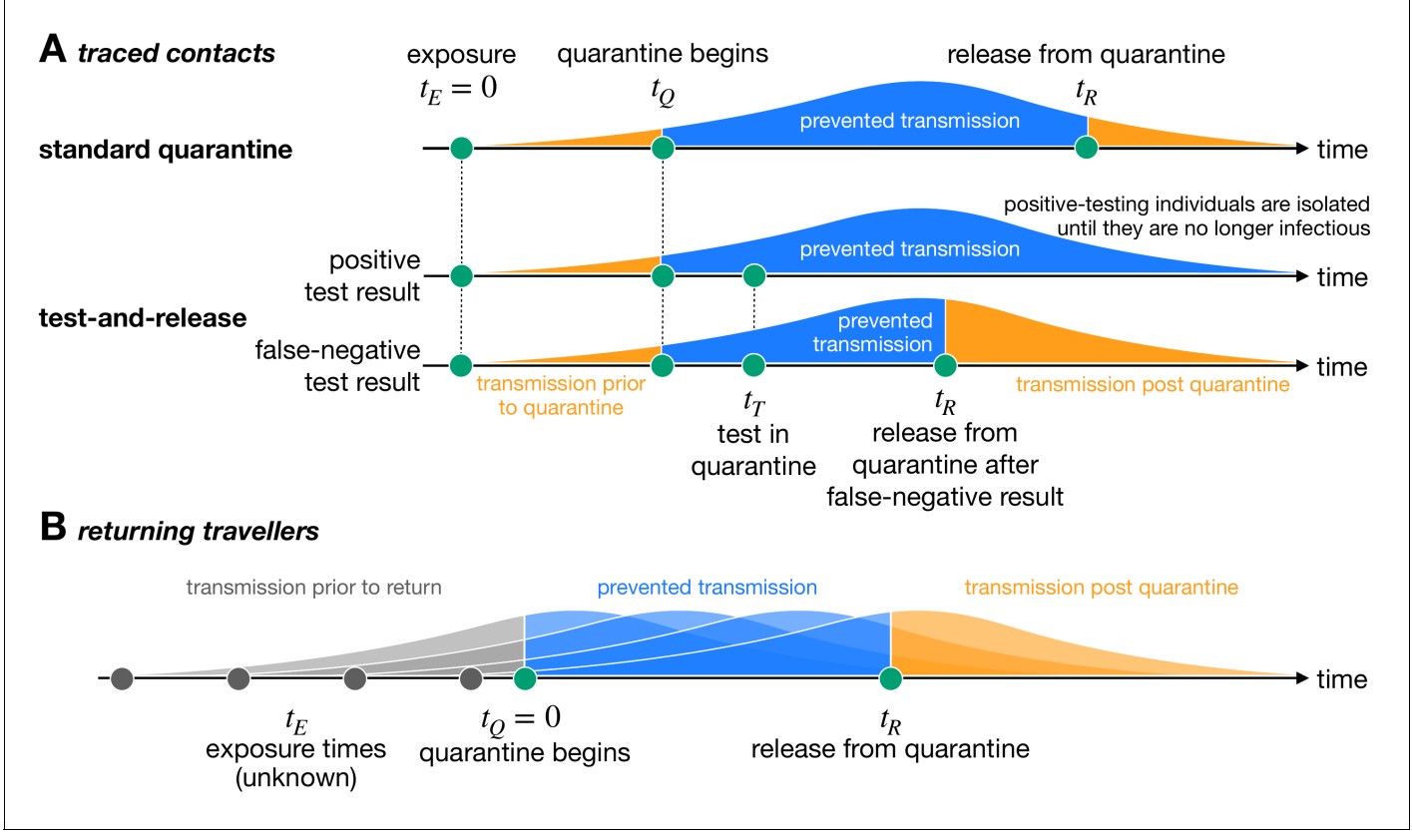

**Figure 1.** Quantifying the impact of quarantine using a mathematical model. Here the y-axis represents the probability of transmission. These infectivity curves are a schematic representation of the generation time distribution shown in *Figure 1—figure supplement 1*. (A) Traced contacts are exposed to an infector at a known time $t_E = 0$ and then enter quarantine at time $t_Q$. Some transmission can occur prior to quarantine. Under the standard quarantine protocol, the contact is quarantined until time $t_R$, and no transmission is assumed to occur during this time. The area under the infectivity curve between $t_Q$ and $t_R$ (blue) is the fraction of transmission that is prevented by quarantine. Transmission can occur after the individual leaves quarantine. Under the test-and-release protocol, quarantined individuals are tested at time $t_T$ and released at time $t_R$ if they receive a negative test result. Otherwise the individual is isolated until they are no longer infectious. The probability that an infected individual returns a false-negative test result, and therefore is prematurely released, depends on the timing of the test relative to infection ($t_T - t_E$) (*Kucirka et al., 2020*). (B) For returning travellers, the time of exposure $t_E$ is unknown and we assume that infection could have occurred on any day of the trip. The travellers enter quarantine immediately upon return at time $t_Q = 0$, and then leave quarantine at time $t_R$ under the standard quarantine protocol. Test-and-release quarantine proceeds as in panel A.

The online version of this article includes the following source data and figure supplement(s) for figure 1:

**Figure supplement 1.** Infection timing distributions.

**Figure supplement 1—source data 1.** Infection timing distributions.

total transmission probability (i.e. the area under the curve) that lies between $t_Q$ and $t_R$ (**Figure 1**). We refer to this fraction of prevented transmission as quarantine efficacy and is defined in **Equation (1)** in 'Materials and methods'. Unless otherwise stated, we assume that adherence to quarantine is 100%.

Under the standard quarantine strategy, all potentially exposed individuals are quarantined for the same duration. An alternative approach is the test-and-release strategy, which uses virological testing during quarantine to release individuals with a negative test result earlier. Individuals with a positive test result are isolated until they are no longer infectious. The timing of the test ($t_T$) is important due to the substantial false-negative rate of the RT-PCR test in the early stages of infection (**Kucirka et al., 2020**). A false-negative test result would release an infected individual into the community prematurely, leading to further transmission (**Figure 1A**). In this case, quarantine efficacy is defined as the expected fraction of transmission that is prevented by quarantine across false-negative and positive testing individuals, as defined in **Equation (2)** in 'Materials and methods'.

As well as the epidemiological benefit of quarantine (i.e. the fraction of transmission prevented by quarantining an infected individual), we can also quantify the economic and societal costs in terms of the expected number of person-days spent in quarantine. We can then define the utility of a quarantine strategy as the ratio between the quarantine efficacy and the average time spent in quarantine, that is, the transmission prevented per day spent in quarantine, as defined in **Equation (4)** in 'Materials and methods'. This utility measure is dependent on the fraction of individuals in quarantine that are infected. This definition of utility should be considered as an example of such a utility function, but this may not be the best way to quantify quarantine utility.

Details of the calculations used can be found in 'Materials and methods'. Further extensions to the model, including the role of reinforced hygiene measures, asymptomatic infections, and adherence to quarantine, are described in Appendix 1.

## Quarantining traced contacts of confirmed SARS-CoV-2 cases

Traced contacts have a known (last) time of exposure to a confirmed case. There is usually a delay between this exposure time and the start of quarantine. Under the standard quarantine protocol, traced contacts are released from quarantine once a number of days have passed after the last exposure time. In Switzerland, for example, quarantine lasts until 10 days after the last exposure.

Any shortening of a traced contact's quarantine duration will lead to an increase in transmission from that individual if they are infected, but the degree of increase depends on the extent of the shortening. The expected onward transmission that is prevented by quarantine shows the diminishing return of increasing the quarantine duration (black line in **Figure 2A**). Increasing quarantine duration beyond 10 days shows almost no additional benefit (**Figure 2—figure supplement 1A**): the standard quarantine protocol (here with a 3-day delay between exposure and the start of quarantine) can maximally prevent 90.8% [95% CI: 79.6%,97.6%] of onward transmission from an infected traced contact, while release on day 10 prevents 90.1% [CI: 76.0%,97.5%].

The maximum attainable prevention also applies to the test-and-release strategy: the onward transmission prevented under a test-and-release strategy will always be below this level (coloured lines in **Figure 2A**). This is because of the chance of prematurely releasing an infectious individual who received a false-negative test result. On the other hand, it is always better to test a person prior to release from quarantine so that individuals with asymptomatic and pre-symptomatic infections can be detected and prevented from being released. Hence, these scenarios provide upper and lower bounds for the efficacy of the test-and-release strategy. The fraction of transmission that is prevented increases if we test later in quarantine because we not only increase the duration of quarantine but also reduce the false-negative probability.

The delay between testing and release from quarantine can have a substantial effect on the efficacy. Current laboratory-based RT-PCR tests have a typical turnaround of 24–48 hr (**Quilty et al., 2021**). This delay is primarily operational, and so could be reduced by increasing test throughput. There are also rapid antigen-detection tests, which can provide same-day results, but with lower sensitivity and specificity than RT-PCR tests (**Guglielmi, 2020**). Here we assume that tests have the same sensitivity and specificity regardless of the delay to result. Compared to a test with 2-day delay until result, we observe that using a rapid test with same-day release can reduce the quarantine duration of individuals with a negative test result by 1 day while maintaining the same efficacy (**Figure 2A**): the extra accuracy gained by waiting one extra day until testing balances the increased

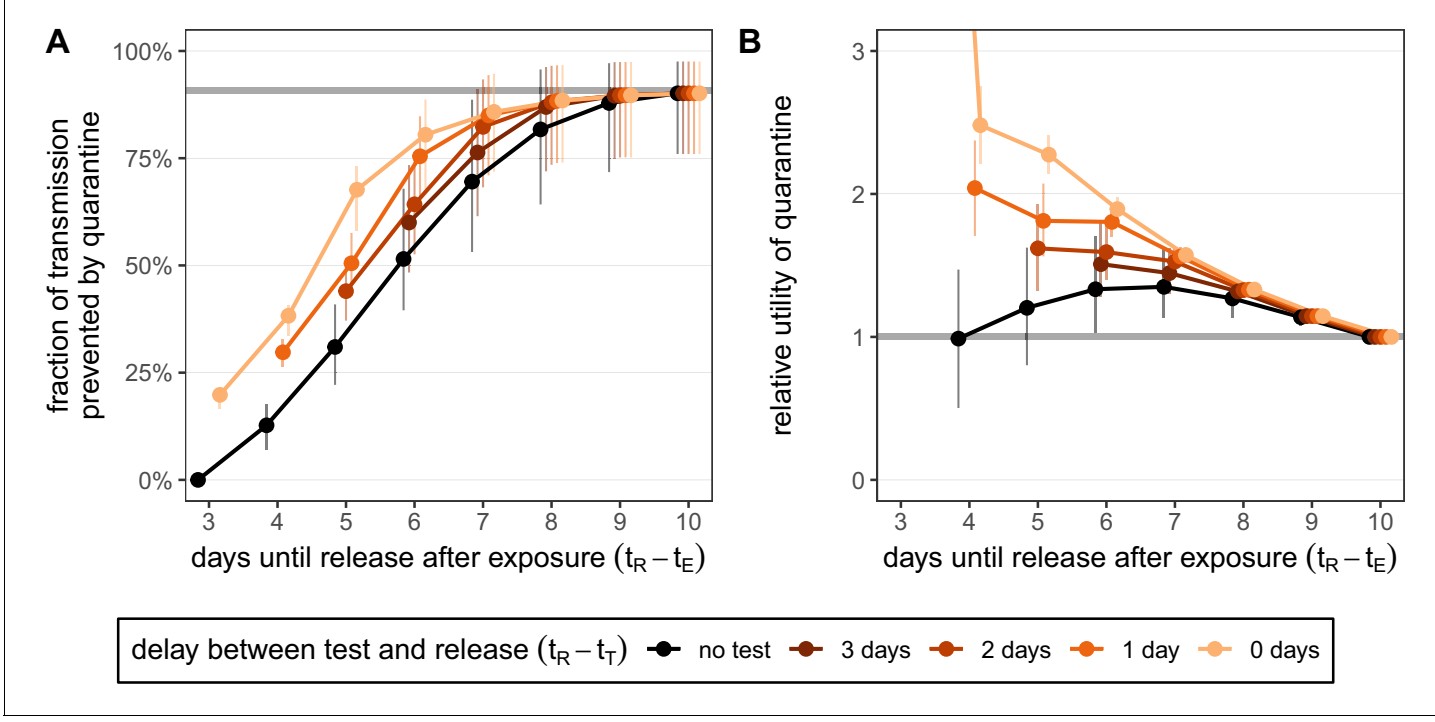

**Figure 2.** Quantifying the impact of quarantine for traced contacts. (**A**) The fraction of transmission that is prevented by quarantining an infected contact. Quarantine begins at time $t_Q = 3$ after exposure at time $t_E = 0$, that is, there is a 3-day delay between exposure and the start of quarantine. Under the standard quarantine protocol (black), individuals are released without being tested [**Equation (1)**]. The test-and-release protocol (colours) requires a negative test result before early release, otherwise individuals remain isolated until they are no longer infectious (day 10) [**Equation (2)**]. Colour intensity represents the delay between test and release (from 0 to 3 days). The grey line represents the maximum attainable prevention by increasing the time of release while keeping $t_Q = 3$ fixed. (**B**) The relative utility of the quarantine scenarios in A compared to the standard protocol 10-day quarantine [**Equation (6)**]. Utility is defined as the fraction of transmission prevented per day spent in quarantine. The grey line represents equal utilities (relative utility of 1). We assume that the fraction of individuals in quarantine that are infected is 10%, and that there are no false-positive test results. Error bars reflect the uncertainty in the generation time distribution.

The online version of this article includes the following source data and figure supplement(s) for figure 2:

**Source data 1.** Fraction of transmission prevented by quarantine (contacts; test-and-release).

**Source data 2.** Relative utility of quarantine (contacts; test-and-release).

**Figure supplement 1.** Quantifying the effect of duration and delay for the standard quarantine protocol (no test) for traced contacts.

**Figure supplement 1—source data 1.** Fraction of transmission prevented by quarantine (contacts; no test).

**Figure supplement 1—source data 2.** Relative utility of quarantine (contacts; no test).

**Figure supplement 2.** Quantifying the impact of quarantine and reinforced hygiene measures for traced contacts.

**Figure supplement 2—source data 1.** Fraction of transmission prevented by quarantine (contacts; reinforced hygiene).

**Figure supplement 2—source data 2.** Relative utility of quarantine (contacts; reinforced hygiene).

transmission caused by reducing the duration. For example, a rapid test on day 6 has roughly the same efficacy (80.5% [CI: 67.9%,88.7%]) as testing on day 5 and releasing on day 7 (82.3% [CI: 68.2%,93.4%]) while shortening the quarantine duration of individuals with a negative test result from 7 to 6 days.

In *Figure 2* we have assumed a fixed delay of 3 days between exposure and the start of quarantine. Shortening this delay increases the maximum efficacy of quarantine because pre-quarantine transmission is reduced (*Figure 2—figure supplement 1A*). If the duration of quarantine is longer than 10 days, then little can be gained in terms of prevention by quarantining for longer, but reducing the delay between exposure and quarantine does lead to increased efficacy.

Note that we have assumed that the contact was infected at the last time of exposure. If there have been multiple contacts between them and the index case, then transmission may have occurred earlier and we would overestimate the efficacy of quarantine.

For the standard quarantine strategy, the duration of quarantine is independent of whether individuals in quarantine are infected. Therefore, the utility of the standard quarantine strategy (i.e. the ratio of efficacy to duration) is directly proportional to the fraction of individuals in quarantine that are infected. By comparing two different standard quarantine strategies through their relative utility (i.e. the ratio of the utilities), we can eliminate the dependence on the fraction of infecteds in quarantine (see 'Materials and methods'). Therefore, the argument that we should shorten quarantine because of the low probability of quarantined individuals being infected is misguided in this situation. By calculating the relative utility for the standard quarantine strategy compared to the baseline 10-day quarantine, we observe that there is a quarantine strategy (release after 7 days) which maximises the ratio between the fraction of transmission prevented and the number of days spent in quarantine (black line in *Figure 2B*). The optimal strategy lies between 6 and 8 days if we vary the delay between exposure and the start of quarantine (*Figure 2—figure supplement 1B*).

Under the test-and-release quarantine protocol, the average time spent in quarantine is dependent on the fraction of infecteds in quarantine; only the infected individuals can test positive and face a longer period of isolation (i.e. we assume there are no false-positive test results). Hence the utility of the test-and-release strategy, as well as the relative utility of test-and-release compared to the standard quarantine protocol, is dependent on the fraction of individuals in quarantine that are infected. In *Figure 2B*, we fix the fraction of infecteds in quarantine to 10%. By calculating the relative utility for the test-and-release quarantine strategies shown in *Figure 2A* compared to the baseline 10-day quarantine, we see that testing-and-releasing before day 10 always increases the utility (*Figure 2B*). Testing on day 5 and releasing test-negative individuals on day 7 has a relative utility of 1.53 [CI: 1.45,1.62] compared to a standard 10-day quarantine. Reducing the delay between test and result leads to a corresponding increase in utility: a rapid test (zero delay between test and result) on day 6 has a relative utility of 1.90 [CI: 1.83,1.98] for an almost equivalent efficacy.

In *Figure 2*, we have made the following assumptions: (i) individuals released from quarantine have – in the post-quarantine phase – the same transmission probability as individuals who were not quarantined; (ii) adherence to quarantine is 100%; and (iii) the transmission prevented by quarantine for cases who develop symptoms is attributed to quarantine. We now relax these assumptions to assess their impact on quarantine efficacy.

Reinforced prevention measures post-quarantine, where individuals who are released from quarantine must adhere to strict hygiene and social distancing protocols until 10 days after exposure have passed, can reduce post-quarantine transmission. Considering a 50% reduction of post-quarantine transmission leads to large increases in both efficacy and utility for early testing strategies, but with diminishing returns as the release date is increased towards day 10 (*Figure 2—figure supplement 2*; see 'Appendix 1: Reinforced prevention measures after early release'). Note that we assume no contribution to the number of days spent in quarantine in the utility function due to mask wearing and social distancing in the post-release phase.

Adherence to quarantine is unlikely to be 100% and could depend on the proposed duration of quarantine. For simplicity we treat adherence to quarantine as a binary variable: a fraction of individuals adhere to quarantine completely for the proposed duration, while the remaining fraction do not undergo any quarantine. We now ask: by how much would the fraction of those who adhere to quarantine have to increase to maintain the efficacy of quarantine if the duration is shortened? In the absence of testing during quarantine, shortening from 10 to 5 days would require almost three times as many individuals to adhere to the quarantine guidelines in order to maintain the same overall efficacy (relative adherence 2.90 [CI: 2.15,4.36]; black line in *Figure 3A*). This threefold increase is not possible if adherence to the 10-day strategy is already above 33% as the maximum adherence cannot exceed 100%; the required increase in adherence grows rapidly as quarantine is shortened and soon becomes infeasible. Hence the argument of shortening quarantine to increase adherence is of limited use. Shortening to 7 days (without testing) may be effective provided that adherence can increase by 30% (relative adherence 1.30 [CI: 1.08,1.55]). Under the test-and-release strategy, however, the efficacy of the standard 10-day quarantine can be matched with release on day 5 or 6 if adherence is also increased by 30%. Releasing earlier than day 5 would seemingly be infeasible given the sharp increase in adherence required.

As a final consideration, we note that our quantification of the fraction of transmission prevented by quarantine is more relevant to individuals with persistently asymptomatic SARS-CoV-2 infection than to those who develop symptoms during quarantine and are subsequently isolated. If

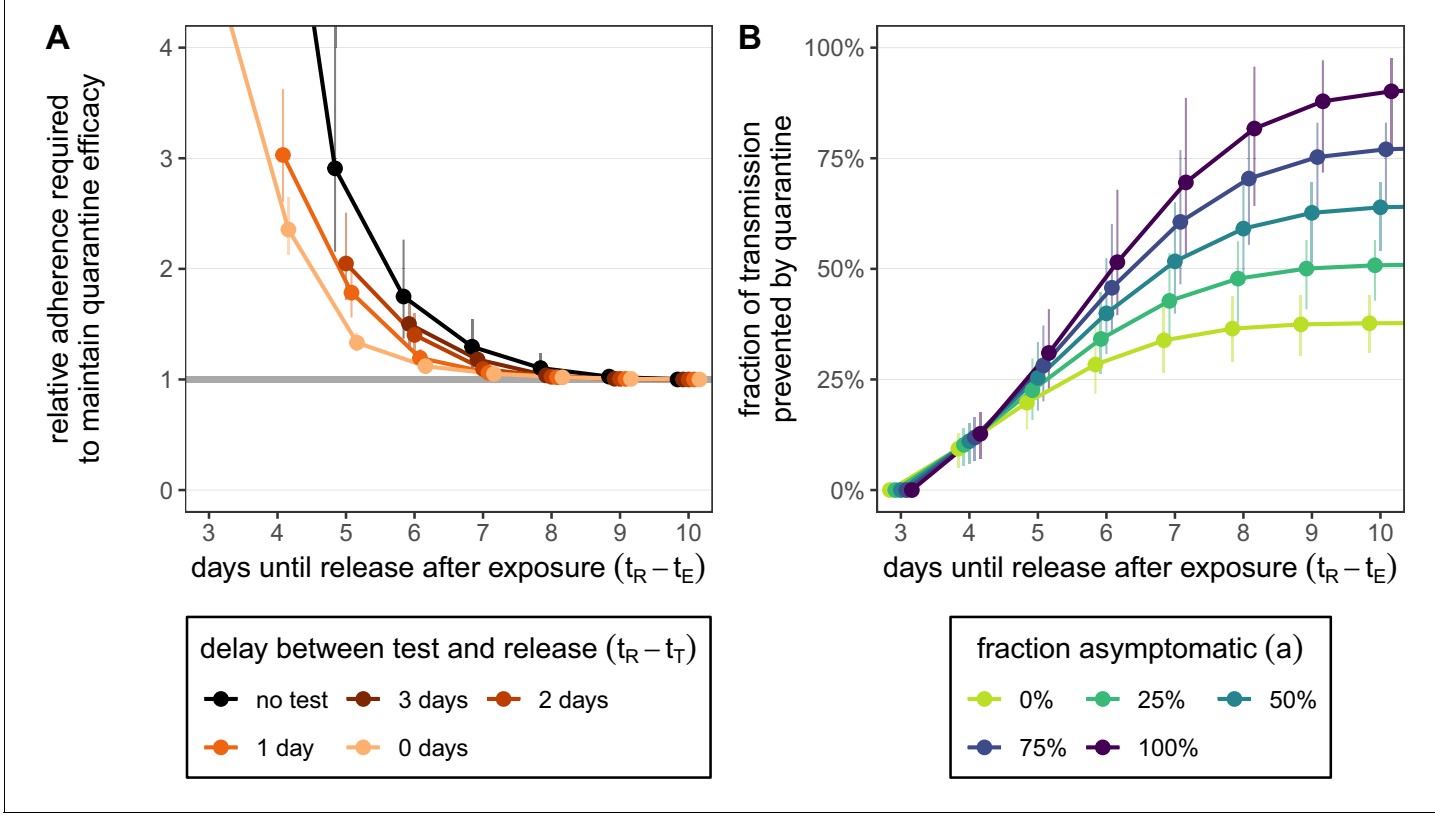

**Figure 3.** How adherence and symptoms affect quarantine efficacy for traced contacts. (A) The fold-change in adherence to a new quarantine strategy that is required to maintain efficacy of the baseline 10-day standard strategy. Quarantine strategies are the same as in *Figure 2* (standard = black, test-and-release = colours). The grey line represents equal adherence (relative adherence of 1). (B) The impact of symptomatic cases on the fraction of total onward transmission per infected traced contact that is prevented by standard (no test) quarantine [*Equation (A9)*]. We assume that symptomatic individuals will immediately self-isolate at symptom onset. The time of symptom onset is determined by the incubation period distribution (see *Figure 1—figure supplement 1D*). The curve for 100% asymptomatic cases corresponds to the black curve in *Figure 2A*. As in *Figure 2*, we fix the time of exposure at $t_E = 0$ and the time of entering quarantine at $t_Q = 3$ days. Error bars reflect the uncertainty in the generation time distribution. The online version of this article includes the following source data and figure supplement(s) for figure 3:

**Source data 1.** Relative adherence (contacts; test-and-release).
**Source data 2.** Role of asymptomatic cases (contacts; zero delay).
**Figure supplement 1.** How the delay between symptom onset and self-isolation affects quarantine efficacy for traced contacts.
**Figure supplement 1—source data 1.** Role of asymptomatic cases (contacts; changing delay).

symptomatic cases go into isolation once symptoms appear, then quarantine has no further impact on transmission after symptom onset as these cases would anyway be isolated. To account for this, we can modify the model such that cases are removed from the infectious pool upon symptom onset (see Appendix 1). For example, in a fully asymptomatic population a 10-day quarantine can prevent 90.1% [CI: 76.0%,97.5%] of transmission. However, if 25% of cases are asymptomatic, then only 50.8% [CI: 42.8%,56.5%] of transmission is prevented by quarantine, while 39.3% is prevented by the self-isolation of symptomatic cases (*Figure 3B*). We assume that self-isolation occurs immediately after symptom onset, but any delay between symptom onset and self-isolation would mean that more transmission is prevented by quarantine (*Figure 3—figure supplement 1*). The fraction of transmission prevented by quarantine is an increasing function of the fraction of asymptomatic cases (*Figure 3B*). This means that we likely overestimate the efficacy of quarantine as we are also counting transmission that could be prevented by isolation following symptom onset. Furthermore, we have assumed that the false-negative rate is the same between symptomatic and asymptomatic cases. If the test is less sensitive (higher false-negative probability) for asymptomatic cases, then quarantine efficacy would be further reduced.

## Quarantining returning travellers

The rules for whether travellers returning from abroad are quarantined are frequently changed according to the epidemiological scenario in the travel destination and/or in the home country. A high risk of infection while abroad due to high prevalence, or the possibility of returning with a new virological variant, can lead to the imposition or reinstatement of quarantine measures (*Russell et al., 2021*). Countries that have already eliminated the infection may be even stricter in their quarantine approach to prevent new community-transmission clusters from being seeded. Here we do not discuss these scenarios or the concept of relative risk, we simply quantify how effective quarantine strategies would be at preventing transmission if the returning traveller was infected while abroad. Should quarantine rules be instated or modified, these results can help determine the optimal quarantine duration and/or testing strategy.

The timing of infection of a traveller during a trip abroad is generally unknown. We assume that infection could have happened on each day of the trip with equal probability. Quarantine begins immediately upon return, which we refer to as day 0, and lasts for a number of days (e.g. currently 10 days in Switzerland) from this timepoint (*Figure 1B*). We consider the fraction of *local* transmission that is prevented by quarantine. That is, the fraction of the transmission that could occur in the local country that is prevented by quarantine [*Equation (8)*]. For a 7-day trip, as in *Figure 4*, the maximum transmission that could occur in the local country is 73.3% [CI: 65.7%,80.3%]. The remaining infectivity potential was already used up before arrival.

A standard (no test) 10-day quarantine will prevent 99.9% [CI: 98.0%,100.0%] of local transmission if the individual was infected during a 7-day trip (*Figure 4A*). There is little benefit to gain by increasing the duration of quarantine beyond 10 days. On the other hand, standard quarantine efficacy decreases quickly as the duration is shortened.

The test-and-release strategy can improve the efficacy of shorter-duration quarantines. Testing on day 5 and releasing on day 7 (to account for test processing delays) performs similarly to 10-day quarantine, preventing 98.5% [CI: 95.5%,99.6%] of local transmission (*Figure 4A*). Testing and releasing on day 6 (i.e. no delay between test and result) still prevents 97.8% [CI: 94.4%,99.0%] of local transmission. Hence, if the rapid test has the same sensitivity and specificity as the laboratory-based RT-PCR test, then the duration of quarantine of individuals with a negative test result can be shortened by 1 day with minimal loss in efficacy compared to a test with a 48 hr turnaround.

The timing of the test can have a significant impact on prevented transmission; late testing reduces the false-negative probability but increases the stay in quarantine. An important consequence of this is that testing on arrival is a poor strategy for limiting transmission: testing and releasing on day 0 would prevent only 35.2% [CI: 35.1%,35.3%] of local transmission, while testing on arrival and releasing after 2 days prevents only 54.1% [CI: 49.5%,59.4%]. As was the case for the traced contacts, the fraction of local transmission prevented by standard quarantine bounds the efficacy of the test-and-release quarantine strategy from below (*Figure 4A*).

We again measure the utility of quarantine by calculating the efficacy (local transmission prevented across all individuals in quarantine, assuming 100% adherence) per day spent in quarantine, and then comparing these utilities for different quarantine strategies to the utility of the standard 10-day quarantine through the relative utility (*Figure 4B*).

In the absence of testing, the duration of quarantine, and hence the relative utility, is independent of the fraction of individuals in quarantine that are infected. For travellers returning from a 7-day trip, the relative utility is a decreasing function of quarantine duration (black line in *Figure 4B*). The maximum utility strategy would then be to shorten quarantine as much as possible.

As was the case for traced contacts, under the test-and-release quarantine protocol the average time spent in quarantine, the utility, and the relative utility compared to the standard 10-day quarantine will depend on the fraction of individuals in quarantine that are infected. This fraction may change depending on disease prevalence at the travel destination and the duration of travel. For example, the infected fraction of travellers returning from a long stay in a high-risk country is likely to be higher than the infected fraction of travellers returning from a short stay to a low-risk country. In *Figure 4B*, we keep this fraction fixed at 10%. Early testing greatly reduces the average duration of quarantine and hence leads to increased utility despite the low fraction of transmission that is prevented (coloured lines in *Figure 4B*).

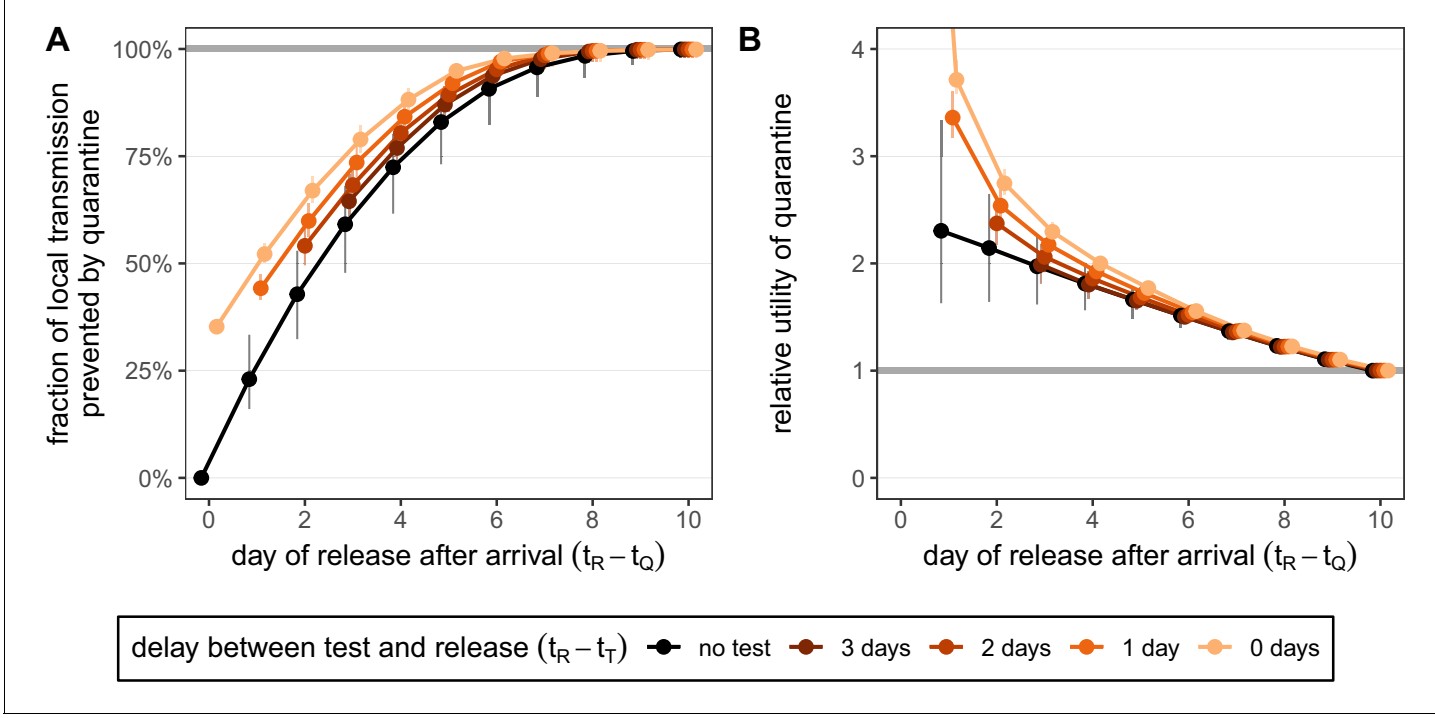

**Figure 4.** Quantifying the impact of quarantine for returning travellers. (A) The fraction of *local* transmission that is prevented by quarantining an infected traveller returning from a 7-day trip. Quarantine begins upon return at time $t_Q = 0$, and we assume that exposure could have occurred at any time during the trip, that is, $-7 \leq t_E \leq 0$. Under the standard quarantine protocol (black), individuals are released without being tested [*Equation (9)*]. The test-and-release protocol (colours) requires a negative test result before early release, otherwise individuals remain isolated until they are no longer infectious (day 10). Colour intensity represents the delay between test and release (from 0 to 3 days). While extended quarantine can prevent 100% of *local* transmission (grey line), this represents 73.3% [CI: 65.7%,80.3%] of the *total* transmission potential (see *Figure 4—figure supplement 1A*). The remaining transmission occurred before arrival. (B) The relative utility of the quarantine scenarios in A compared to the standard protocol 10-day quarantine [*Equation 6*]. Utility is defined as the *local* fraction of transmission that is prevented per day spent in quarantine. The grey line represents equal utilities (relative utility of 1). We assume that the fraction of individuals in quarantine that are infected is 10%, and that there are no false-positive test results. Error bars reflect the uncertainty in the generation time distribution.

The online version of this article includes the following source data and figure supplement(s) for figure 4:

**Source data 1.** Fraction of transmission prevented by quarantine (travellers; local; test-and-release).

**Source data 2.** Relative utility of quarantine (travellers; local; test-and-release).

**Figure supplement 1.** Quantifying the effect of travel duration and quarantine duration for the standard quarantine protocol (no test) for returning travellers.

**Figure supplement 1—source data 1.** Fraction of transmission prevented by quarantine (travellers; total; no test).

**Figure supplement 1—source data 2.** Relative utility of quarantine (travellers; total; no test).

**Figure supplement 1—source data 3.** Fraction of transmission prevented by quarantine (travellers; local; no test).

**Figure supplement 1—source data 4.** Relative utility of quarantine (travellers; local; no test).

**Figure supplement 2.** Quantifying the impact of quarantine and reinforced hygiene measures for returning travellers.

**Figure supplement 2—source data 1.** Fraction of transmission prevented by quarantine (travellers; local; reinforced hygiene).

**Figure supplement 2—source data 2.** Relative utility of quarantine (travellers; local; reinforced hygiene).

**Figure supplement 3.** How adherence and symptoms affect quarantine efficacy for returning travellers.

**Figure supplement 3—source data 1.** Relative adherence (travellers; local; test-and-release).

**Figure supplement 3—source data 2.** Role of asymptomatic cases (travellers; local).

The average quarantine duration increases linearly with the fraction of infecteds in quarantine [*Equation (3)* in 'Materials and methods']. The ratio of quarantine efficacy to the average quarantine duration will also increase, such that quarantine is of higher utility if the fraction of infecteds is higher. However, the relative utility of test-and-release quarantine compared to the standard 10-day protocol will decrease and approach 1 as the fraction of infecteds increases. Hence, if the disease

prevalence among those returning from travel abroad is high, then test-and-release may not bring substantial benefits over the standard 10-day protocol.

Our assumption that infection occurs with uniform probability across each day of a trip leads to some interesting results. Returning travellers that have been infected on a short journey will have, on average, used up less of their infectivity potential by the time they return than a traveller who was infected on a long journey. Hence, the *total* transmission that can be prevented by a long quarantine period (e.g. 10 days) upon arrival is greater for short trips (*Figure 4—figure supplement 1A*). When considering the fraction of *local* transmission that can be prevented by quarantine, then shorter quarantine durations have a greater impact on long than short trips (*Figure 4—figure supplement 1C*). Again, this is because, on average, the traveller on a long trip would have been exposed earlier and they will be infectious for a shorter time period after arrival.

If an individual traveller is to be quarantined, then the optimum duration of quarantine, based on our metric of utility, would depend on the duration of their travel, with shorter journeys requiring longer quarantine (*Figure 4—figure supplement 1B, D*). This might be counterintuitive because individuals who have been on longer journeys to high-risk countries have a higher probability of being infected. The absolute utility (transmission prevented by quarantine across all individuals in quarantine divided by the average quarantine duration) of quarantining such individuals could indeed be higher than for individuals returning from shorter journeys. However, here, we are not considering the question of whether to quarantine or not, but we are assuming that the individual is quarantined and are trying to optimise the duration of quarantine in response to the expected infection dynamics.

We observe an almost-linear response between quarantine duration and the relative utility of the standard (no test) quarantine: for every day that quarantine is shortened, we see the same additive increase in relative utility (black line in *Figure 4B*). This almost-linear response is coincidental to the 7-day trip duration: longer or shorter trips show non-linear responses (*Figure 4—figure supplement 1D*). Trips shorter than 7 days have a maximum relative utility of between 4 and 7 days, while trips longer than 7 days have maximum utility for maximally shortened quarantine durations.

Enforcing additional hygiene and social distancing guidelines post-quarantine can increase both efficacy and utility, but with diminishing returns as the release date is increased (*Figure 4—figure supplement 2*).

As discussed for traced contacts, the loss of efficacy due to shortening quarantine could be offset by increasing quarantine adherence. Shortening from 10 to 5 days would require adherence to increase by 20% (relative adherence 1.20 [CI: 1.12,1.35]) in order to maintain the same overall efficacy (*Figure 4—figure supplement 3A*). With test-and-release this required increase in adherence is even smaller. We note that the change in adherence required to balance a change in efficacy for shortened quarantine durations is dependent on the travel duration, with short travel durations requiring a greater increase in adherence compared with longer travel durations.

## Discussion

Quarantine is one of the most important measures in controlling the ongoing SARS-CoV-2 epidemic due to the large fraction of pre-symptomatic and asymptomatic transmission. A quarantine period of 10 days from exposure, as currently implemented in Switzerland, is long enough to prevent almost all onward transmission from infected contacts of confirmed cases from the point of entering quarantine: increasing the duration of quarantine beyond 10 days has no extra benefit. Reducing the delay to quarantining individuals increases the fraction of total transmission that is preventable. The same 10-day quarantine duration will prevent almost all local onward transmission from infected travel returners from the time of arrival, independent of their travel duration.

Any decrease in the duration of quarantine of an infected individual will result in increased onward transmission. Furthermore, our analyses suggest that this increase in transmission cannot realistically be compensated by increased adherence for significantly shortened quarantine (fewer than 5 days). However, there are diminishing returns for each day that we add to quarantine: increasing the duration from 10 days has a negligible effect in terms of reduced transmission. One therefore has to assess how much human cost, measured in terms of days spent in quarantine, we are willing to spend to prevent disease transmission. By comparing the ratio of prevented transmission to quarantine duration, we have shown that maximal utility strategies can exist. This ratio is maximised for

quarantine durations of 6–8 days after exposure for traced contacts, and potentially less for returning travellers depending on their duration of travel. Importantly, under this metric the fraction of individuals in quarantine that are infected does not affect the optimal duration of quarantine. Therefore, the argument that we should shorten quarantine because of the low probability of being infected is misguided under our definition of utility and in the absence of testing during quarantine.

A test-and-release strategy will lead to a lower average quarantine duration across infected and non-infected individuals. However, due to the considerable false-negative probability of the RT-PCR test (*Kucirka et al., 2020*), this strategy also leads to increased transmission as infectious individuals are prematurely released. Nevertheless, test-and-release strategies prevent more transmission than releasing without testing, and hence test-and-release increases the utility of quarantine. Reducing the delay between test and result leads to further reduced transmission and increased utility, and reinforcing individual prevention measures after release is also effective for short quarantine periods.

The ratio of transmission prevented versus days spent in quarantine is only one possible definition of utility. Defining the appropriate function is ultimately a policy question: the economic, societal, and individual costs are likely a function of the number of days spent in quarantine, but we cannot determine the shape of this function. Furthermore, the local epidemiological situation will dictate which metric of quarantine efficacy is to be optimised. In situations where the goal is to prevent the (re)introduction of SARS-CoV-2, one should focus on maximising the reduction of transmission (and hence minimising the transmission risk). If the virus is already endemic, then considering the trade-off between transmission reduction and quarantine duration could determine the optimum strategy. Another perspective is that the utility of preventing transmission is crucially dependent on whether it brings the effective reproductive number under 1.

Ultimately, bringing the reproductive number below 1 requires a combination of effective measures including isolation, physical distancing, hygiene, contact tracing, and quarantine (*Kucharski et al., 2020*). Effective quarantine is only possible in the presence of efficient contact tracing to find the potentially exposed individuals in a short time, as well as surveillance of disease prevalence to identify high-risk travel. Further reducing the time taken to quarantine a contact after exposure and reducing the delay between test and result will allow average quarantine durations to be shorter, which increases the benefit-to-cost ratio of quarantine.

The scenarios of returning travellers and traced contacts of confirmed SARS-CoV-2 cases differ in the probability of having been exposed and infected and on the information available about the likely window of exposure. The impact of quarantining returning travellers depends on the duration of travel and whether we consider the local prevention of transmission or the total transmission prevented by quarantine. However, a single test done immediately after return can only prevent a small fraction of the transmission from a returning traveller because of the false-negative rate of the RT-PCR test early in infection. Therefore testing should be postponed until as late as possible, and utilising rapid tests could be crucial if their performance characteristics are acceptable. This same principle also applies to traced contacts. Our findings are aligned with those of two recent simulation studies which estimate the role that quarantine plays in limiting transmission from returning travellers (*Clifford, 2020*) and from traced contacts (*Quilty et al., 2021*).

Our results are based on the latest estimates of the generation time distribution of COVID-19 (*Ferretti et al., 2020b*). Potential limitations to our approach could be that these distributions may change throughout the epidemic, particularly depending on how people respond to symptoms (*Ali et al., 2020*). Furthermore, these distributions, and also the test sensitivity profile, could be different between persistently asymptomatic and symptomatic individuals (*Buitrago-Garcia et al., 2020*), which ultimately lead to an overestimation of how much transmission is prevented by quarantine. In addition, we have primarily assumed that symptom onset during quarantine has no impact on quarantine efficacy. However, this symptomatic transmission should not be counted towards the efficacy of quarantine as the infected individual should already self-isolate after symptom onset. We have quantified this effect and have shown that this assumption leads us to overestimate quarantine efficacy.

For travellers, another consideration is that lengthy quarantine is seen as a deterrent to travel to high-risk areas (*IATA, 2020*). Any shortening of quarantine may lead to an increase in travel volume, potentially leading to a compounded increase in disease transmission.

In the absence of empirical data about the effectiveness of different durations of quarantine, mathematical modelling can be used objectively to explore the fraction of onward transmission by

infected contacts or returning travellers that can be prevented. However, determining the optimal quarantine strategy to implement depends on the impact that shortening the duration of quarantine has on individuals, society, and the economy versus how much weight is assigned to a consequential increase in transmission. Both the individual, societal, and economic impact, as well as the weight of transmission increase, will have to be considered based on the current epidemiological situation. We have shown that there are quarantine strategies based on a test-and-release protocol that, from an epidemiological viewpoint, perform almost as well as the standard 10-day quarantine, but with a lower cost in terms of person-days spent in quarantine. This applies to both travellers and contacts, but the specifics depend on the context.

## Materials and methods

### Quantifying the benefit of quarantine

For an infected individual who was exposed at time $t_E$, the fraction of transmission that is prevented by the standard quarantine strategy is given by the area under the generation time distribution, $q(t)$ (*Figure 1—figure supplement 1B*), between the times at which the individual enters ($t_Q$) and leaves ($t_R$) quarantine (*Grantz et al., 2020*), that is,

$$F_{\mathrm{qs}}(t_E, t_Q, t_R) = \int_{t_Q}^{t_R} \mathrm{d}t\, q(t - t_E).$$

(1)

The duration of time that the individual spends in quarantine is then $D_{\mathrm{qs}} = t_R - t_Q$.

The test-and-release strategy uses virological testing during quarantine to release individuals with a negative test result and to place those with a positive test result into isolation. As illustrated in *Figure 1A*, test is issued at time $t_T \geq t_Q$. If the test is negative, the individual is released when the test result arrives at time $t_R$. Otherwise, the individual is isolated until they are no longer infectious. One challenge with this strategy is the high probability of a false-negative RT-PCR test result (i.e. an infectious individual is prematurely released into the community). As reported by *Kucirka et al., 2020*, the false-negative rate is 100% on days 0 and 1 post-infection, falling to 67% (day 4), 38% (day 5), 25% (day 6), 21% (day 7), 20% (day 8), and 21% (day 9), before rising to 66% on day 21. We use linear interpolation and label this function $f(t)$, the false-negative probability on day $t$ after infection. The fraction of transmission prevented by quarantining an infected individual under the test-and-release strategy is

$$F_{\mathrm{qtr}}(t_E, t_Q, t_T, t_R) = f(t_T - t_E) \int_{t_Q}^{t_R} \mathrm{d}t\, q(t - t_E) + [1 - f(t_T - t_E)] \int_{t_Q}^{t_{\mathrm{end}}} \mathrm{d}t\, q(t - t_E),$$

(2)

where the first term captures the fraction of individuals who receive a false-negative test result and are released at time $t_R$, and the second term captures individuals who return a positive test and are subsequently isolated until they are no longer infectious at time $t_{\mathrm{end}}$. A further challenge with this false-negative rate is that it was calculated by *Kucirka et al., 2020* from symptomatic cases only. Here we assume that this test sensitivity profile also applies to asymptomatic cases.

Quarantine is applied pre-emptively, such that we do not know the infection status of individuals when they enter quarantine. If only a fraction $s$ of the individuals that are quarantined are infected, then the average reduction in transmission across all individuals in quarantine is $sF$, where $F$ is the fraction of transmission prevented when an infected individual is quarantined [i.e. *Equation (1)* or *(2)*]. For the standard quarantine protocol, the average number of days spent in quarantine is independent of $s$: all individuals are quarantined for the same duration. However, under the test-and-release protocol, only the individuals who are actually infected can test positive and remain isolated after $t_R$. All non-infected individuals $(1 - s)$ will receive a negative test result and are released at time $t_R$. Among the infected individuals in quarantine ($s$), a fraction $f(t_T - t_E)$ will receive a false-negative test result and will be released at time $t_R$, while the remaining fraction $[1 - f(t_T - t_E)]$ will receive a positive test result and are isolated until they are no longer infectious. Hence the average number of days spent in quarantine for test-and-release is

$$D_{\mathrm{qtr}} = (1-s)(t_R - t_Q) + s\big[f(t_T - t_E)(t_R - t_Q) + [1-f(t_T - t_E)](t_{\mathrm{end}} - t_Q)\big]$$
$$= (t_R - t_Q) + s[1 - f(t_T - t_E)](t_{\mathrm{end}} - t_R), \tag{3}$$

where $s[1 - f(t_T - t_E)]$ is the fraction of quarantined individuals who return a positive test result. We see that the average test-and-release quarantine duration increases linearly with the fraction of individuals in quarantine that are infected ($s$).

Model parameters and timepoints are summarised in *Table 1*.

## Transmission reduction versus days spent in quarantine

One possible metric to relate the effectiveness of quarantine to its negative impact on society is to consider the ratio between the amount of overall transmission prevented and the number of person-days spent in quarantine. We refer to this ratio as the utility of quarantine. Concretely, for an efficacy $F$ [$F_{\mathrm{qs}}$ or $F_{\mathrm{qtr}}$ as defined by *Equation (1) or (2)*, respectively], fraction of individuals in quarantine that are infected $s$, and average time spent in quarantine $D$ ($D_{\mathrm{qs}}$ or $D_{\mathrm{qtr}}$), we define the utility as

$$U(s, F, D) = \frac{sF}{D}. \tag{4}$$

We can then compare the utility of two quarantine strategies by calculating the relative utility, that is, the ratio between the two utilities:

$$\mathrm{RU}(s, F, D, F^*, D^*) = \frac{sF/D}{sF^*/D^*} = \frac{F/D}{F^*/D^*}, \tag{5}$$

where $F$ and $D$ are the efficacy and duration of quarantine of a new strategy, and $F^*$ and $D^*$ are the efficacy and duration of the baseline quarantine strategy to which we compare.

**Table 1.** Summary of terms used in the mathematical model.

| Value | Definition | Notes |
|---|---|---|
| $q(t)$ | Generation time distribution | Weibull distribution: shape = 3.277, scale = 6.127 |
| $t_E$ | Time of exposure | $t_E = 0$ for traced contacts |
| $t_Q$ | Time at which quarantine begins | $t_Q = 0$ for returning travellers |
| $t_R$ | Time of release from quarantine | - |
| $t_T$ | Time of test | - |
| $t_{\mathrm{end}}$ | End of infectiousness | $t_{\mathrm{end}} = t_E + 10$ days |
| $g(t)$ | Incubation period distribution | Meta-log-normal distribution ('Appendix 1: Distribution parameters') |
| $t_S$ | Time of symptom onset | $t_S = t_E +$ incubation period |
| $D_{\mathrm{qs}}$ | Realised average duration of standard quarantine | $D_{\mathrm{qs}} = t_R - t_Q$ |
| $D_{\mathrm{qtr}}$ | Realised average duration of test-and-release quarantine | See *Equation (3)* |
| $F_{\mathrm{qs}}(\cdot)$, $F_{\mathrm{qtr}}(\cdot)$ | Quarantine efficacy; the fraction of transmission prevented by quarantining an infected individual | See *Equations (1) and (2)* |
| $y$ | Duration of travel journey (days) | - |
| $s$ | Fraction of individuals in quarantine that are infected | - |
| $f(t)$ | Probability of returning a false-negative test result if tested $t$ days after exposure | From *Kucirka et al., 2020* |
| $r$ | Reduction of transmission under reinforced prevention measures post-quarantine | - |
| $\alpha(D)$ | Probability to adhere to quarantine of duration $D$ | - |
| $a$ | Fraction of persistently asymptomatic cases | - |
| $\Delta$ | Delay between symptom onset and isolation (days) | See 'Appendix 1: Persistently asymptomatic infections and the role of self-isolation' |

The efficacies $F$ and $F^*$ in *Equation (5)* are independent of fraction of individuals in quarantine that are infected $s$. For the standard quarantine strategy, the durations $D = D_{\mathrm{qs}}$ and $D^* = D^*_{\mathrm{qs}}$ are also independent of $s$, and hence the relative utility of the standard quarantine strategy is independent of $s$. For the test-and-release strategy, however, the duration is a linearly increasing function of $s$ [$D = D_{\mathrm{qtr}}(s)$; *Equation (3)*]. Hence the relative utility of the test-and-release strategy is dependent on $s$:

$$\mathrm{RU}[s, F_{\mathrm{qtr}}, D_{\mathrm{qtr}}(s), F^*_{\mathrm{qs}}, D^*_{\mathrm{qs}}] = \frac{F_{\mathrm{qtr}}/D_{\mathrm{qtr}}(s)}{F^*_{\mathrm{qs}}/D^*}. \tag{6}$$

In Appendix 1 we show that the relative utility of the test-and-release quarantine strategy is a decreasing function of $s$.

## Traced contacts versus returning travellers

We consider the scenarios of a traced contact and a returning traveller differently because the values of $t_E$, $t_Q$, and $t_R$ are implemented differently in each case.

### Traced contacts

Following a positive test result, a confirmed index case has their recent close contacts traced. From contact tracing interviews, we know the date of last exposure between index case and a contact ($t_E$), which we assume is the time of infection of the contact. The contacts begin quarantine at time $t_Q \geq t_E$. The delay between exposure and the start of quarantine represents the sum of the delay to the index case receiving a positive test result and the further delay to tracing the contacts. Under the standard quarantine protocol, the traced contacts are quarantined for a number of days after their last exposure. For example, in Switzerland quarantine lasts until $t_R = t_E + 10$ days, but may be longer or shorter depending on individual states' regulations. Note that the actual time spent in quarantine is $D_{\mathrm{qs}} = t_R - t_Q$ days, which is typically shorter than 10 days due to the delay between exposure and the start of quarantine. For convenience, we set $t_E = 0$ for the traced contacts, without loss of generality.

### Returning travellers

Unlike traced contacts, we generally do not know when travellers were (potentially) exposed. This means that quarantine starts from the date that they return ($t_Q = 0$) and lasts until time $t_R$ (*Figure 1B*). For simplicity, we assume that a traveller was infected with uniform probability at some time over a travel period of duration $y$ days.

For each possible exposure time $-y \leq t_E \leq 0$ during the trip, we can compute the fraction of transmission prevented using *Equation (1)* and then take the average over the possible exposure times. This represents the average fraction of the *total* transmission potential that is prevented by quarantining this traveller:

$$\overline{F}^{(\mathrm{total})}_{\mathrm{qs}}(y, t_R) = \frac{1}{y+1} \sum_{t_E = -y}^{0} \int_0^{t_R} \mathrm{d}t \, q(t - t_E), \tag{7}$$

where we have used $t_Q = 0$.

For each exposure time $-y \leq t_E \leq 0$, we can also compute the *local* fraction of transmission prevented by quarantine, which is the fraction of transmission prevented by quarantine divided by the maximum amount of transmission that could occur in the local country, that is,

$$F^{(\mathrm{local})}_{\mathrm{qs}}(t_E, t_R) = \frac{\int_0^{t_R} \mathrm{d}t \, q(t - t_E)}{\int_0^{\infty} \mathrm{d}t \, q(t - t_E)}, \tag{8}$$

where we have again used $t_Q = 0$. Taking the average over the possible exposure times $-y \leq t_E \leq 0$, we have

$$\overline{F}_{\mathrm{qs}}^{(\mathrm{local})}(y, t_R) = \frac{1}{y+1} \sum_{t_E=-y}^{0} F_{\mathrm{qs}}^{(\mathrm{local})}(t_E, t_R).$$

(9)

## Interactive app

To complement the results in this paper, and to allow readers to investigate different quarantine scenarios, we have developed an online interactive application. This can be found at https://ibz-shiny.ethz.ch/covidDashboard/quarantine.

## Additional information

### Funding

| Funder | Grant reference number | Author |
|---|---|---|
| H2020 European Research Council | EpiPose 101003688 | Nicola Low |
| Swiss National Science Foundation | 176233 | Nicola Low |
| Swiss National Science Foundation | 176401 | Sebastian Bonhoeffer |

The funders had no role in study design, data collection and interpretation, or the decision to submit the work for publication.

### Author contributions

Peter Ashcroft, Conceptualization, Data curation, Software, Formal analysis, Validation, Investigation, Visualization, Methodology, Writing - original draft, Project administration, Writing - review and editing; Sonja Lehtinen, Conceptualization, Data curation, Formal analysis, Methodology, Writing - original draft, Writing - review and editing; Daniel C Angst, Data curation, Software, Validation, Visualization, Writing - review and editing; Nicola Low, Supervision, Funding acquisition, Writing - review and editing; Sebastian Bonhoeffer, Conceptualization, Formal analysis, Supervision, Funding acquisition, Methodology,Writing - original draft, Project administration, Writing - review and editing

### Author ORCIDs

Peter Ashcroft https://orcid.org/0000-0003-4067-7692
Sonja Lehtinen http://orcid.org/0000-0002-4236-828X
Daniel C Angst https://orcid.org/0000-0002-6512-4595
Nicola Low https://orcid.org/0000-0003-4817-8986
Sebastian Bonhoeffer https://orcid.org/0000-0001-8052-3925

### Decision letter and Author response

Decision letter https://doi.org/10.7554/eLife.63704.sa1
Author response https://doi.org/10.7554/eLife.63704.sa2

## Additional files

### Supplementary files

• Transparent reporting form

### Data availability

All data generated or analysed during this study are included in the manuscript and supporting files. These files are available on github https://github.com/ashcroftp/COVID-quarantine/ (copy archived at https://archive.softwareheritage.org/swh:1:rev:7492ec296dab70e3eda5479fedc56f8310cc4417/) and archived at https://doi.org/10.5281/zenodo.4580232.

The following dataset was generated:

| Author(s) | Year | Dataset title | Dataset URL | Database and Identifier |
|---|---|---|---|---|
| Ashcroft P, Lehtinen S, Angst DC, Low N, Bonhoeffer S | 2021 | ashcroftp/quarantine2020 | https://doi.org/10.5281/zenodo.4580232 | Zenodo, 10.5281/zenodo.4580232 |

The following previously published dataset was used:

| Author(s) | Year | Dataset title | Dataset URL | Database and Identifier |
|---|---|---|---|---|
| Ferretti L | 2020 | Code & data for "The timing of COVID-19 transmission" | https://doi.org/10.5281/zenodo.4033022 | Zenodo, 10.5281/zenodo.4033022 |

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

## Appendix 1

### Utility and relative utility of test-and-release quarantine

From *Equation (4)* of 'Materials and methods', we can write the utility of the test-and-release strategy as

$$U(s, F_{\text{qtr}}, D_{\text{qtr}}(s)) = \frac{sF_{\text{qtr}}}{D_{\text{qtr}}(s)}, \tag{A1}$$

where $F_{\text{qtr}}$ is the quarantine efficacy [*Equation (2)* in 'Materials and methods'], $s$ is the fraction of individuals in quarantine that are infected, and $D_{\text{qtr}}(s)$ is the average time spent in quarantine [*Equation (3)* in 'Materials and methods']. The duration $D_{\text{qtr}}(s)$ is a linear function of $s$, which we can write simply as $D_{\text{qtr}}(s) = ms + b$ [from *Equation 3* in 'Materials and methods', we have $m = [1 - f(t_T - t_E)](t_{\text{end}} - t_R)$ and $b = (t_R - t_Q)$].

We now ask, how does this utility change if we increase $s$? Taking the derivative of *Equation (A1)* with respect to $s$, we recover

$$\frac{\mathrm{d}U}{\mathrm{d}s} = \frac{\mathrm{d}}{\mathrm{d}s}\left(\frac{sF_{\text{qtr}}}{ms + b}\right) = \frac{bF_{\text{qtr}}}{(ms + b)^2} > 0. \tag{A2}$$

Hence, any increase in $s$ leads to an increase in utility.

The relative utility of the test-and-release quarantine strategy to the standard quarantine strategy is defined in *Equation (6)* in 'Materials and methods'. Again taking the derivative with respect to $s$, we recover

$$\frac{\mathrm{d}\,\text{RU}}{\mathrm{d}s} = \frac{\mathrm{d}}{\mathrm{d}s}\left(\frac{F_{\text{qtr}}/(ms + b)}{F_{\text{qs}}^*/D_{\text{qs}}^*}\right) = \frac{-mD_{\text{qs}}^* F_{\text{qtr}}}{F_{\text{qs}}^*(ms + b)^2} < 0. \tag{A3}$$

Hence, any increase in $s$ leads to a decrease in the relative utility of the test-and-release strategy compared to the standard quarantine strategy.

### Reinforced prevention measures after early release

We further consider the possibility that individuals who are released early from quarantine are asked to strengthen hygiene, mask wearing, and social distancing protocols until the end of the infectious period. We assume that these practices reduce transmission by a fraction $r$ such that the onward transmission prevented by quarantining an infected individual and reinforcing hygiene measures is

$$F_{\text{qs}}^{(\text{reduced})}(t_E, t_Q, t_R) = F_{\text{qs}}(t_E, t_Q, t_R) + r \int_{t_R}^{t_{\text{end}}} \mathrm{d}t\, q(t - t_E), \tag{A4}$$

for the standard quarantine protocol, and

$$F_{\text{qtr}}^{(\text{reduced})}(t_E, t_Q, t_T, t_R) = F_{\text{qtr}}(t_E, t_Q, t_T, t_R) + rf(t_T - t_E) \int_{t_R}^{t_{\text{end}}} \mathrm{d}t\, q(t - t_E), \tag{A5}$$

for the test-and-release protocol.

### Adherence to quarantine

For the majority of our results, we have assumed that quarantine is completely adhered too. Because of this assumption we will overestimate the efficacy of quarantine at the level of the population as it is likely that adherence will be less than 100%.

Adherence could be included as a time-varying property of an individual such that the probability that an individual follows the quarantine guidelines is high at the beginning of quarantine, but is waning as the duration spent in quarantine increases. However, for simplicity, we assume that adherence is binary; either an individual completes the full duration of quarantine or they do not enter quarantine at all. We denote the probability to adhere to quarantine as $\alpha(D)$, which depends on the quarantine duration $D$. The average fraction of transmission prevented by (standard)

quarantine is then $s\alpha(t_R - t_Q)F_{qs}(t_E, t_Q, t_R)$, where $s$ is the fraction of individuals in quarantine that are infected, which we assume is independent of quarantine duration, and we have used $D = D_{qs} = t_R - t_Q$.

We do not know the adherence probabilities $\alpha(D)$. However, for two quarantine strategies with release dates $t_R$ and $t_R^*$ to have the same efficacy they must satisfy

$$s\alpha(t_R - t_Q)F_{qs}(t_E, t_Q, t_R) = s\alpha(t_R^* - t_Q)F_{qs}(t_E, t_Q, t_{R^*})$$
$$\Rightarrow \quad \frac{\alpha(t_R - t_Q)}{\alpha(t_R^* - t_Q)} = \frac{F_{qs}(t_E, t_Q, t_R^*)}{F_{qs}(t_E, t_Q, t_R)}. \tag{A6}$$

That is to say, the change in the fraction of transmission prevented by quarantine must be compensated by an inverse change in the adherence: a strategy which prevents half as much transmission as another would require adherence to be doubled to be equally effective. We therefore define the required relative adherence as

$$\mathrm{RA}_{qs}(t_E, t_Q, t_R, t_R^*) = \frac{F_{qs}(t_E, t_Q, t_R^*)}{F_{qs}(t_E, t_Q, t_R)}. \tag{A7}$$

This definition of relative adherence is directly extended to the test-and-release strategy, which we compare to the baseline standard protocol:

$$\mathrm{RA}_{qtr}(t_E, t_Q, t_T, t_R, t_R^*) = \frac{F_{qs}(t_E, t_Q, t_R^*)}{F_{qtr}(t_E, t_Q, t_T, t_R)}. \tag{A8}$$

## Persistently asymptomatic infections and the role of self-isolation

If an individual develops symptoms, is tested, and ultimately tests positive while in quarantine, we can remove them from the infectious pool as they would have to isolate themselves. Importantly, this symptomatic individual would be removed from the infectious pool whether they have been placed in quarantine or not. Therefore, this symptomatic transmission should not be counted towards the efficacy of quarantine.

Let $a$ be the fraction of asymptomatic cases who will be quarantined using the standard strategy from time $t_Q$ until $t_R$. The symptomatic cases (which make up a fraction $1 - a$ of cases) will develop symptoms at time $t_S$, as described by the incubation period distribution shown in *Figure 1—figure supplement 1D*. We assume that the symptomatic cases would be isolated shortly after they develop symptoms at time $t_S + \Delta$, so these individuals are effectively quarantined until time $\min(t_R, t_S + \Delta)$. We further assume equal transmissibility of persistently asymptomatic and symptomatic infections and that both are described by the same generation time distribution. This assumption might be an overestimate as onward transmission from persistently asymptomatic cases is less than onward transmission from symptomatic cases (*Buitrago-Garcia et al., 2020*). For each traced contact who is put into quarantine, the fraction of infections that would be prevented by quarantine is

$$
\begin{aligned}
F_{qs}^{(asymp)}(t_E, t_Q, t_R, a, \Delta) \;=\; & a \int_{t_Q}^{t_R} \mathrm{d}t\, q(t - t_E) \\
& + (1 - a) \int_{t_E}^{\infty} \mathrm{d}t_S\, g(t_S - t_E) \int_{t_Q}^{\min(t_R, t_S + \Delta)} \mathrm{d}t\, q(t - t_E),
\end{aligned} \tag{A9}
$$

where $g(t)$ is the incubation period distribution, and the outer integral over $t_S$ is the averaging over the possible times of symptom onset. Note that this formulation assumes that the timing of onward transmission is independent of the incubation period (see *Ferretti et al., 2020b* and *Lehtinen et al., 2021* for further discussion of this assumption). Unless otherwise stated, we assume $a = 1$ and $\Delta = 0$.

## Confidence intervals

The primary source of uncertainty in the outcomes of this model comes from the generation time distribution, which is inferred from the empirical serial interval distribution combined with the incubation period distribution (*Ferretti et al., 2020b*). Following *Ferretti et al., 2020b*, we use a

likelihood ratio test to extract sample parameter sets for the generation time distribution that lie within the 95% confidence interval.

Concretely, we first identify the parameter set $\hat{\theta}$ for the generation time distribution which maximises the likelihood of observing the empirical serial interval distribution. The likelihood function and fitting process are described in detail by *Ferretti et al., 2020b*. The generation time distribution is described by a Weibull distribution (with $n = 2$ parameters). We then randomly sample the parameter space of the generation time distribution and keep 1000 parameter sets whose likelihood satisfies $\ln \mathcal{L}(\theta) > \ln \mathcal{L}(\hat{\theta}) - \lambda_2/2$, where $\lambda_2$ is the 95% quantile of a $\chi^2$ distribution with $n = 2$ degrees of freedom. These parameter sets, as shown in *Appendix 1—figure 1*, define the 95% confidence interval for the generation time distribution (*Figure 1—figure supplement 1B*).

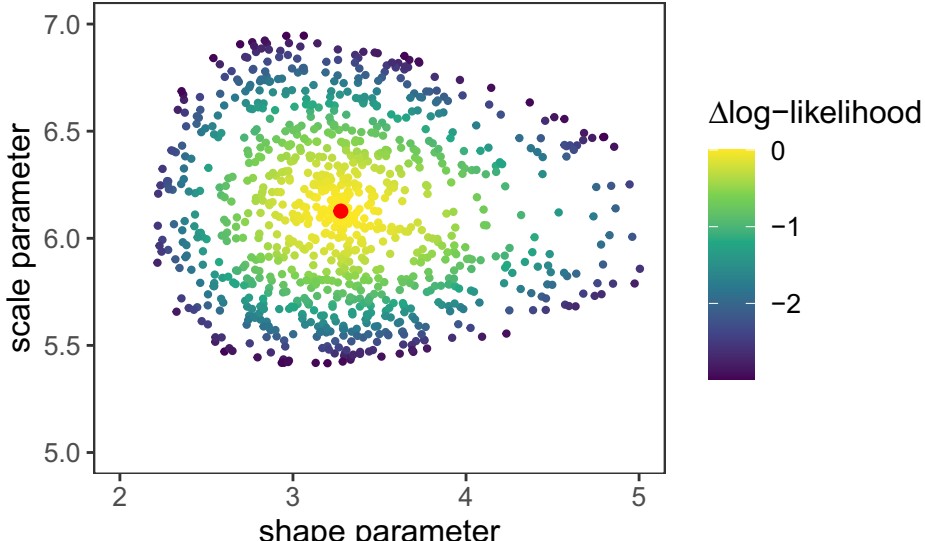

**Appendix 1—figure 1.** Log-likelihood values $[\ln \mathcal{L}(\theta) - \ln \mathcal{L}(\hat{\theta})]$ for random parameter samples of the generation time distribution. These samples define the 95% confidence interval for the generation time distribution parameters. Red dot is the maximum likelihood parameter combination as shown in *Appendix 1—table 1*.

The online version of this article includes the following source data is available for figure 1:

**Appendix 1—figure 1—source data 1.** Generation time distribution parameters versus likelihood.

We then use these sampled parameter sets to calculate quarantine efficacy, and the extrema of these efficacies across all of these parameter sets determines the 95% confidence interval of the efficacy.

## Distribution parameters

The parameters that define the incubation period distribution, generation time distribution, and infectivity profile are shown in *Appendix 1—table 1*.

**Appendix 1—table 1.** Parameters of the distributions used in this work.

The meta-log-normal distribution is the average of seven reported log-normal distributions (*Ferretti et al., 2020b*). The shifted Student's *t* distribution for the infectivity profile is defined in R by dt((x-shift)/scale, df)/scale (*Ferretti et al., 2020b*).

| Distribution | Shape | Parameters | Properties |
|---|---|---|---|
| Incubation period | Meta-log-normal | meanlog = 1.570, sdlog = 0.650 (Bi) | mean = 5.723, sd = 3.450, median = 4.936 |
| | | meanlog = 1.621, sdlog = 0.418 (Lauer) | |
| | | meanlog = 1.434, sdlog = 0.661 (Li) | |
| | | meanlog = 1.611, sdlog = 0.472 (Linton) | |
| | | meanlog = 1.857, sdlog = 0.547 (Ma) | |
| | | meanlog = 1.540, sdlog = 0.470 (Zhang) | |
| | | meanlog = 1.530, sdlog = 0.464 (Jiang) | |
| Generation time | Weibull | shape = 3.277, scale = 6.127 | mean = 5.494, sd = 1.845, median = 5.479 |
| Infectivity profile | Shifted Student's t | shift = -0.078, scale = 1.857, df = 3.345 | mean = -0.042, sd = 2.876, median = -0.078 |

