## [Decision Letter]

**Acceptance summary:**

This work provides a detailed look into the potential benefits of a test and release quarantine strategy. By using quantitative models of the timing of quarantine, testing, release and transmission, the authors are able to show that testing people during quarantine, and releasing them after a negative test, could provide similar efficacy in terms of reducing transmission, while shortening the burden on quarantines individuals. This is relevant to policy decisions as balances are sought between efficacy and societal cost of quarantine.

**Decision letter after peer review:**

Thank you for submitting your article "Quantifying the impact of quarantine duration on COVID-19 transmission" for consideration by *eLife*. Your article has been reviewed by four peer reviewers, one of whom is a guest Reviewing Editor and the evaluation has been overseen by Miles Davenport as the Senior Editor. The following individual involved in review of your submission has agreed to reveal their identity: Mirjam Kretzschmar (Reviewer #2).

The Reviewing Editor has drafted this decision to help you prepare a revised submission.

In this paper the authors compare a standard quarantine scenario with a test and release quarantine scenario, and look at the efficacy and utility of each. This is a timely and important analysis, however as it is presented it is very technical, and not well explained for a general audience. It should therefore be revised, and technical details placed in an appendix.

The authors have successfully summarised both the advantages and disadvantages of a test and release quarantine strategy, as well as quarantine for different durations, and have shown that in some circumstances this may be preferable to a standard strategy.

At the moment, the paper does not clearly explain some of the more technical aspects, despite having many equations. It would be better served by explaining well, in plain language the approaches taken, and leaving the mathematics for a technical appendix.

Essential Revisions:

1) Equations must be tidied and there must be a consistency of notation. Many of them should be removed to a technical appendix, and explained in a more intuitive way in the text so the manuscript can be read by a more general audience.

2) Figures should be combined and edited to make paper accessible to a more general audience.

3) Assumptions made must be contextualised and explained. Specifically (but not exclusively):

a) Why do the authors repeatedly state that someone with a positive test is released on day tR+ when in actual fact they would be in isolation – please amend text and clarify.

b) Discuss how the local community transmission level impacts on the conclusions that can be drawn about returned travellers.

c) Consider how the likelihood that symptomatic individuals (not in already in quarantine) will isolate impacts the conclusions.

Reviewer #1:

This paper is well thought through and presents a very nice analysis of the benefits (and disadvantages) of a test and release quarantine strategy, as well as considering different durations of quarantine. I have some concerns however, with the presentation and the assumptions made, as well as the context in which these strategies are undertaken.

In regards to presentation, there are many mathematical equations presented throughout the manuscript, however in many cases these are confusing. As an example, Equations 2, 3 and 4 have no parameters for the function F, and do not contain an n in them, but F is repeatedly used with the parameter n. The relationship between n, t_R_, t_Q_, t_E_ etc should be explained and the functions parameterised more clearly.

Additionally, there are 5 result figures presented, which are difficult to interpret. The authors should consider alternative ways of representing their results to make them easier to understand for a non-technical reader.

Regarding assumptions made, the comments in section “Persistently asymptomatic infections” that state that symptomatic individuals would be removed from the population (N.B. this should read infectious pool rather than population) regardless of their quarantine status are only true if ALL people self-isolate on the day of symptom onset. Since this is unlikely to be the case, there will only be a probability of isolation occurring x days after symptom onset, p(x). Presumably p(x) will increase after (a) testing and (b) a positive result, but we cannot assume it will be 1 on the day of symptom onset. This needs to be addressed.

In section “Adherence to quarantine” a function α(n) is introduced, but α(n) is not shown in the paper, and it is not clear whether subsequent calculations include this α(n) or not. This must be clarified. Also, the authors do not consider the possibility that α may wane over time (i.e. for a fixed duration of quarantine, people may quarantine effectively for the first half and but less effectively for the second half). This should be either commented on, or addressed.

It is not clear from the text what is meant by "the test-and-release strategy always performs better than not testing if the release time is the same as the quarantine duration". This must be clarified, and can be done by stating "It will always be better to test a person prior to release from quarantine, as that way asymptomatic and pre-symptomatic infections are more likely to be detected and prevented from being released".

Finally, the context of this manuscript is not made clear. Early on, the authors state that quarantine of returning travellers occurs when "they have returned from recent travel to a high-risk area with levels of community transmission that are higher than in the home country" however in many countries that is not the strategy that is being implemented. Many countries are requiring quarantine after travel to countries with similar or slightly less community transmission. Therefore, the assumption that returning travellers must have contracted their infection overseas is incorrect. This only works in regions where there is little community transmission in the home country. Therefore, if this is to be presented as an argument, these results need to be stratified by cases where the visited country has either much more, similar or much less community transmission than the home country.

Reviewer #2:

In this manuscript the authors develop a computational approach to determine the impact of test sensitivity and duration of quarantine on onward transmission of COVID-19. Based on the generation time distribution, the authors estimate what part of onward transmission can be prevented, given test sensitivity per day since exposure, and delays in testing. They also introduce a utility measure for quarantine, that takes into account the average time spent in quarantine. Overall this is a well written paper with a clear approach and useful results on various quarantining strategies.

There are a few ways in which this paper could be made more accessible for the reader. The notation is mathematical, and notation of variables not always very intuitive. As many quantities have similar names, it would help the reader to have a table with explanation of the variable names, so that one can look back quickly while reading to check how a variable was defined. Alternatively (of in addition), the graphs in Figure 1 could be extended to include more variables, for example a similar graph could be made for the traced contacts, and for returning travellers.

Reviewer #3:

Quantifying the impact of quarantine is an important topic, where the literature is currently lacking. This piece is very timely and tackles some very important questions in the space. At the moment, I feel this paper is somewhat lacking and requires some revision. I think the biggest issue is that it seems to be tailored for the situation in Switzerland. In itself, that's fine, but if the paper stays that way, the title should be altered to specify "in Switzerland", and the Introduction should give some more detail about the local situation. If the authors made that change, I have only a few technical queries that need to be addressed. I will also give some comments and suggestions about the differences between quarantine for close contacts compared to returned travellers, and what should be done to make the analysis more useful globally.

1) I'm concerned that the calculations in this paper are incorrect. Specifically, I'm looking the second paragraph in subsection “Quantifying the benefit of quarantine” and Equation 1. If q(_t_) is the probability density of the generation time (and generation time is the time from exposure until the person becomes infectious), then the integral in Equation 1 is the probability that the person becomes infectious while in quarantine. I don't see how this is equal to the fraction of transmission prevented. I thought you would need the function to be the probability that someone is infectious at time t? Which I think is the convolution of Figure 2A and 2B. I am not too concerned about this, as if it is an error it shouldn't be hard to fix.

2) The other technical issue is about returned travellers, and how the trip duration affects the transmission reduction through quarantine. I'm not convinced by the argument that short trips need longer quarantine. I think the calculations are ok, and the result comes from the assumptions themselves. The authors are assuming that a traveller contracts covid while away, so on a 1 day trip, they must have caught it recently, and therefore need more time for symptoms to develop and to test positive. However, I don't think the authors should rule out a traveller catching covid before leaving, and becoming infectious after returning. I understand that the authors want to split the paper up into locally acquired cases, and international, but I don't think it works. If the situation was one where the country had no covid, then this claim about short trips needing longer quarantine makes sense. But in general it's a more nuanced question and certainly doesn't seem to fit Switzerland. I guess it could also work if you assume all local transmissions can be contact-traced, but I don't think the authors claimed this either.

3) On to the more general point about applicability globally, which is specific to returning travellers (apart from the notes above, I think the close contact part of this work look good). I think the authors have missed explaining the decision context clearly (where the decision is about length of quarantine), and how differing values, objectives and system states affect things (see Baker et al., 2020 for a general discussion on decision making). The point I want to get to is, what is the objective of quarantine for returned travellers? The paper focusses on reduction of spread, which seems reasonable for countries with ongoing community transmission. However, it would be important to be clear about how much transmission in the country is coming through airports, relative to the community spread.

There are many countries and jurisdictions that have either no ongoing transmission, or very well contained clusters. Quarantine in these areas is completely different, as the aim isn't about getting some percentage reduction is transmission. Instead, the aim is to have 100% reduction, and the metric of interest is the probability that an infection escapes quarantine and seeds a cluster of cases in the community. I think many people would be very interested in an analysis that looked at quarantine length and testing strategy in this context, to get towards a trade-off between length of stay, costs and importation risk.

Overall, I think this paper has made some good steps. However, the message needs to be refined, and the context of the paper needs to be clarified. I think quarantine is a very important topic, and in its current state, the manuscript is only applicable to regions with active transmission. I think the current scope is acceptable at this journal, but it needs to be clear that it is aiming for that. Even without analysis suited to low/no prevalence scenarios, at a minimum there should be some discussion about how the local epidemiological situation is driving the results and the analysis, and how quarantine may need to be approached differently elsewhere.

I wish the authors all the best in their revisions. As I said, it's an important topic and there needs to be more literature about it. I would happily review this paper again. I also commend the authors for making everything available on Github. This was useful during my review.

References:

Baker, C.M., Campbell, P.T., Chades, I., Dean, A.J., Hester, S.M., Holden, M.H., McCaw, J.M., McVernon, J., Moss, R., Shearer, F.M. & Possingham, H.P. (2020). From climate change to pandemics: decision science can help scientists have impact. ArXiv200713261 Phys.

Reviewer #4:

Ashcroft et al. discussed the appropriate duration of quarantine for travellers and close-contacts by estimating the fraction of transmission that can be avoided under a range of scenario and their relative utility. The study also explored the impact of testing, reinforced hygiene, adherence, and symptom presentation on transmission. This is an important and policy relevant topic. The article is nicely written, and I only have a few comments.

Cases included in Kucirka et al. are all symptomatic, and they back calculated time from exposure assuming incubation period of 5 days. The authors here seem to assume that the sensitivity of the RT-PCR test found in symptomatic individuals can be applied to asymptomatic individuals. Please state this assumption (if that's correct) and discuss how this assumption impacts the outcome.

Figure 4 and other figures, can the authors state how the upper and lower bounds are estimated?

---

## [Author Response]

Essential Revisions:1) Equations must be tidied and there must be a consistency of notation. Many of them should be removed to a technical appendix, and explained in a more intuitive way in the text so the manuscript can be read by a more general audience.

We have followed the Editor’s recommendation and have removed all equations from the main text. To explain our approach in a more intuitive way, we have reworked Figure 1 which now graphically captures our methodology. We have added a verbal description of the model at the start of the Results section to complement this schematic.

All notation has now been streamlined by removing a number of terms (*n*,*x*,*t_R_*±,∆*_Q_*,∆*_T_*).

In the main text we now only refer to the timepoints of exposure (*t_E_*), quarantine

entry (*t_Q_*), testing (*t_T_*), and release (*t_R_*), which are introduced in the revised Figure 1.

All equations describing the fraction of transmission prevented, utility, and relative utility in Materials and methods and in the Appendix now have explicit and consistent arguments based on these timepoints.

Finally, we have added a glossary of terms (Table 1 in Materials and methods) which can be used as a reference guide for our notation.

2) Figures should be combined and edited to make paper accessible to a more general audience.

Following our revisions, the main text now only contains four figures: one schematic figure and three figures containing the main results. We have combined the figures for standard quarantine and test-and-release for traced contacts into a single figure (now Figure 2). This now allows for direct comparison between the alternate quarantine strategies. We have repeated this for the returning travellers too (now Figure 4). We have modified the result figure that shows the impact of adherence (now Figure 3A) to also include test-and-release, which as pointed out by reviewer 2 (comment 2.9) contains an important result relating to minimum quarantine durations.

In all figures we have tried to reduce the information content and increase clarity, which we hope makes our results more accessible. As a consequence, we have added extra supplemental figures, all of which are referred to in the main text. Further details are described in the response to reviewer 1.

3) Assumptions made must be contextualised and explained. Specifically (but not exclusively):a) Why do the authors repeatedly state that someone with a positive test is released on day tR+ when in actual fact they would be in isolation – please amend text and clarify.b) Discuss how the local community transmission level impacts on the conclusions that can be drawn about returned travellers.c) Consider how the likelihood that symptomatic individuals (not in already in quarantine) will isolate impacts the conclusions.

We thank the Editor for pointing this out. While streamlining our notation, we have removed the timepoints *t_R_*±. Now we state explicitly that the infected individual who receives a positive test result while in quarantine is isolated until they are no longer infectious. In the Model description section of Results, we write: “Under the standard quarantine strategy, all potentially-exposed individuals are quarantined for the same duration. An alternative approach is the test-and-release strategy, which uses virological testing during quarantine to release individuals with a negative test result earlier. Individuals with a positive test result are isolated until they are no longer infectious.”. This point is reiterated in the Materials and methods section: “If the test is negative, the individual is released when the test result arrives at time *t_R_*. Otherwise, the individual is isolated until they are no longer infectious.”.

Based on the generation time distribution, the infectious period lasts for 10 days after exposure, so we fix the release time to day 10, as noted in the captions of Figures 2 and 4, and in the glossary in Table 1.

b) This point was raised by reviewer 1 and reviewer 3. In this paper we focus on quantifying how much transmission is prevented by different quarantine durations and test-and-release strategies. This quantification is not affected by the relative risk of local community transmission versus import from travellers. What determines the optimal quarantine strategy ultimately depends on the local epidemiological situation and the goal of the authorities: Preventing (re)introduction of the virus would correspond maximising the transmission prevented by quarantine. However, if the virus is already endemic then we could be looking for trade-offs between transmission prevention and quarantine duration.

We have now expanded these points regarding what defines the optimal quarantine strategy at the beginning of the Results section for returning travellers: “The rules for whether travellers returning from abroad are quarantined are frequently changed according to the epidemiological scenario in the travel destination and/or in the home country. A high risk of infection while abroad due to high prevalence, or the possibility of returning with a new virological variant, can lead to the imposition or reinstatement of quarantine measures. Countries that have already eliminated the infection may be even stricter in their quarantine approach to prevent new community-transmission clusters from being seeded. Here we do not discuss these scenarios or the concept of relative risk, we simply quantify how effective quarantine strategies would be at preventing transmission if the turning traveller was infected while abroad. Should quarantine rules be instated or modified, these results can help determine the optimal quarantine duration and/or testing strategy.”.

c) Reviewers 1 and 2 have brought up points surrounding the role that symptom onset plays in our results. In addition to the existing results in Figures 3B and 4—figure supplement 3B, we have now included an additional supplementary figure (Figure 3—figure supplement 1) describing how delays between symptom onset and isolation further affect the results. We have clarified that we assume self-isolation occurs immediately in Figures 3B and 4—figure supplement 3B. We have also improved the description and equations describing these results in the Appendix. Finally, we have added this point as a limitation in our Discussion: “In addition, we have primarily assumed that symptom onset during quarantine has no impact on quarantine efficacy. However, this symptomatic transmission should not be counted towards the efficacy of quarantine as the infected individual should already self-isolate after symptom onset. We have quantified this effect and have shown that this assumption leads us to overestimate quarantine efficacy.”.

We hope it is now clear that self-isolation after symptom onset is only considered in Figures 3B, 3—figure supplement 1, and 4—figure supplement 3B, and that its impact is to reduce the fraction of transmission that is prevented by quarantine.

Reviewer #1:This paper is well thought through and presents a very nice analysis of the benefits (and disadvantages) of a test and release quarantine strategy, as well as considering different durations of quarantine. I have some concerns however, with the presentation and the assumptions made, as well as the context in which these strategies are undertaken.In regards to presentation, there are many mathematical equations presented throughout the manuscript, however in many cases these are confusing. As an example, Equations 2, 3 and 4 have no parameters for the function F, and do not contain an n in them, but F is repeatedly used with the parameter n. The relationship between n, t_R_, t_Q_, t_E_ etc should be explained and the functions parameterised more clearly.

We apologise for the confusion caused by our notation. Upon revision, we have streamlined our notation to focus only on the defined timepoints *t_E_*, *t_Q_*, *t_T_*, and *t_R_*(respectively time of exposure, start of quarantine, time of test, and time of release). These timepoints are now annotated in the new schematic figure (Figure 1). The equations (now located in Materials and methods and the Appendix) now have explicit arguments based on these timepoints. Furthermore, we have removed a number of terms (*n*,*x*,*t_R_*±,∆*_Q_*,∆*_T_*) that can be described by the previously-mentioned timepoints. We hope with the introduction of the schematic figure and the simplified notation that any confusion/ambiguity has been removed.

Additionally, there are 5 result figures presented, which are difficult to interpret. The authors should consider alternative ways of representing their results to make them easier to understand for a non-technical reader.

In the revised manuscript we now have only four figures: one schematic and three containing the results. We believe this improves the accessibility and readability of our study. To make our result figures more interpretable, we have tried to reduce the information content of the figures, while making it easier to make comparisons between the standard quarantine strategy and test-and-release.

To this end, in the main results for traced contacts we have focussed on only one delay-to-quarantine value (3 days), and combined Figures 3 and 4 into a single figure (now Figure 2) containing the no-test scenario and test-and-release. The previous Figure 3 (varying the delay-to-quarantine) has moved to Figure 2—figure supplement 1, and so have the dashed lines from previous Figure 4 (now Figure 2—figure supplement 2) for the reduced post-quarantine transmission scenario.

Adherence and the role of symptoms are still discussed in Figure 3 (previously Figure 5), but with the same quarantine strategies (standard + test-and-release) as presented in Figure 2 and rescaled axes for visual clarity.

In the same fashion as Figures 3 and 4, we have combined Figures 6 and 7 for the returning travellers (now Figure 4). We now focus only on a travel duration of 7 days, and move Figure 6 with the impact of travel duration to Figure 4—figure supplement 1.

Regarding assumptions made, the comments in section “Persistently asymptomatic infections” that state that symptomatic individuals would be removed from the population (N.B. this should read infectious pool rather than population) regardless of their quarantine status are only true if ALL people self-isolate on the day of symptom onset. Since this is unlikely to be the case, there will only be a probability of isolation occurring x days after symptom onset, p(x). Presumably p(x) will increase after (a) testing and (b) a positive result, but we cannot assume it will be 1 on the day of symptom onset. This needs to be addressed.

In response to this point, we have investigated the impact of adding a delay between symptom onset and self-isolation (Figure 3—figure supplement 1). For the main results we stick with self-isolation immediately after symptom onset (zero delay). This zero-delay scenario is informative as it corresponds to the minimum efficacy of quarantine; any transmission that would occur between symptom onset and self-isolation would be prevented by quarantine, leading to an increase in efficacy. We now clarify this assumption in the Results section: “We assume self-isolation occurs immediately after symptom onset, but any delay here would mean more transmission is prevented by quarantine (Figure 3—figure supplement 1).”. The corresponding equation is written and described in the Appendix section “Persistently asymptomatic infections and the role of self-isolation”.

Thank you for pointing out our poor phrasing, we have replaced “remove them from the population” with “remove the symptomatic cases from the infectious pool”, and likewise in the Appendix.

In section “Adherence to quarantine” a function α(n) is introduced, but α(n) is not shown in the paper, and it is not clear whether subsequent calculations include this α(n) or not. This must be clarified. Also, the authors do not consider the possibility that α may wane over time (i.e. for a fixed duration of quarantine, people may quarantine effectively for the first half and but less effectively for the second half). This should be either commented on, or addressed.

We acknowledge that the adherence function *α* was not sufficiently or consistently described in the original submission. In response, we have removed *α* completely from the main text. This function now only features in the glossary table and in the Appendix (section “Adherence to quarantine”), where we define the relative adherence measure [Appendix 1—equation (3)].

We now explicitly state that we have assumed 100% adherence in the Model description: “Unless otherwise stated, we assume that adherence to quarantine is 100%.”; and that this assumption is relaxed for Figure 3A: “In Figure 2 have made the following assumptions: (i) individuals released from quarantine have – in the post-quarantine phase – the same transmission probability as individuals who were not quarantined; (ii) quarantine is 100% adhered to; and (iii) the transmission prevented by quarantine for cases who develop symptoms is attributed to quarantine. We now relax these assumptions to assess their impact on quarantine efficacy.”. In the Results text before Figure 3, we introduce the concept of adherence and clarify that we treat it as a binary variable: “Adherence to quarantine is unlikely to be 100% and could depend on the proposed duration of quarantine. For simplicity we treat adherence to quarantine as a binary variable: a fraction of individuals adhere to quarantine completely for the proposed duration, while the remaining fraction do not undergo any quarantine. We now ask: by how much would the fraction of those who adhere to quarantine have to increase to maintain the efficacy of quarantine if the duration is shortened?”. This is expanded in the Appendix: “Adherence could be included as a time-varying property of an individual, such that the probability that an individual follows the quarantine guidelines is high at the beginning of quarantine, but is waning as the duration spent in quarantine increases. However, for simplicity, we assume adherence is binary; either an individual completes the full duration of quarantine, or they do not enter quarantine at all.”.

We believe that with our verbal description of adherence and the assumptions, and the expanded Appendix (section “Adherence to quarantine”), that the confusion introduced around adherence has been clarified.

It is not clear from the text what is meant by "the test-and-release strategy always performs better than not testing if the release time is the same as the quarantine duration". This must be clarified, and can be done by stating "It will always be better to test a person prior to release from quarantine, as that way asymptomatic and pre-symptomatic infections are more likely to be detected and prevented from being released".

We thank the reviewer for their suggested explanation, which we have incorporated into the text: “On the other hand, it is always better to test a person prior to release from quarantine, so that individuals with asymptomatic and pre-symptomatic infections can be detected and prevented from being released.”.

Finally, the context of this manuscript is not made clear. Early on, the authors state that quarantine of returning travellers occurs when "they have returned from recent travel to a high-risk area with levels of community transmission that are higher than in the home country" however in many countries that is not the strategy that is being implemented. Many countries are requiring quarantine after travel to countries with similar or slightly less community transmission. Therefore, the assumption that returning travellers MUST have contracted their infection overseas is incorrect. This only works in regions where there is little community transmission in the home country. Therefore, if this is to be presented as an argument, these results need to be stratified by cases where the visited country has either much more, similar or much less community transmission than the home country.

We thank the reviewer for this comment and for helping us to sharpen the point of our manuscript. In our work we follow the guidelines of the World Health Organtion (WHO) who say: “There are two scenarios in which quarantine may be implemented: (1) for travellers from areas with community transmission and (2) for contacts of known cases.” (WHO, 2020). They then go on to say: “If Member States choose to implement quarantine measures for travellers on arrival at their destination, they should do so based on a risk assessment and consideration of local circumstances.” (WHO, 2020). In our work we are not considering the question of whether quarantine should be imposed on returning travellers or not: we do not consider the risk of introduction or the relative risk of infection abroad versus at home. Such an analysis was recently published by Russell et al., 2020. In our work we want to quantify how quarantine duration (and testing) impacts the amount of transmission if quarantine is implemented for returning travellers. We then assume that the returning traveller was infected during their trip, and that potential transmission is prevented by quarantine. The relative risk of infection could feature in a measure of utility, but that is beyond the scope of our paper.

In response, we have generalised our introductory statement about the circumstances under which returning travellers should be quarantined, using the guide lines from the WHO: “Individuals are generally placed into quarantine for one of two reasons: either they have been identified as a recent close contact of a confirmed SARS-CoV-2 case by contact tracing, or they have returned from recent travel to an area with community transmission that has been assessed to pose significant epidemiological risk (WHO, 2020).”.

We have expanded on this further in Results section “Quarantining returning travellers”: “The rules for whether travellers returning from abroad are quarantined are frequently changed according to the epidemiological scenario in the travel destination and/or in the home country. A high risk of infection while abroad due to high prevalence, or the possibility of returning with a new virological variant, can lead to the imposition or reinstatement of quarantine measures. Countries that have already eradicated the disease may be even stricter in their quarantine approach to prevent new community-transmission clusters from being seeded. Here we do not discuss these scenarios or the concept of relative risk, but we simply quantify how effective quarantine strategies would be at preventing transmission if the returning traveller was infected while abroad. Should quarantine rules be instated or modified, these results can help determine the optimal quarantine duration and/or testing strategy.”.

We hope these modifications have now clarified the message of our paper.

Reviewer #2:In this manuscript the authors develop a computational approach to determine the impact of test sensitivity and duration of quarantine on onward transmission of COVID-19. Based on the generation time distribution, the authors estimate what part of onward transmission can be prevented, given test sensitivity per day since exposure, and delays in testing. They also introduce a utility measure for quarantine, that takes into account the average time spent in quarantine. Overall this is a well written paper with a clear approach and useful results on various quarantining strategies.There are a few ways in which this paper could be made more accessible for the reader. The notation is mathematical, and notation of variables not always very intuitive. As many quantities have similar names, it would help the reader to have a table with explanation of the variable names, so that one can look back quickly while reading to check how a variable was defined. Alternatively (of in addition), the graphs in Figure 1 could be extended to include more variables, for example a similar graph could be made for the traced contacts, and for returning travellers.

We thank the reviewer for their positive comments about the project, and for their constructive suggestions. We believe these changes have greatly improved the quality and accessibility of the manuscript.

We have decided to follow both advices suggested here. We have redrawn Figure 1 to include more clearly the relevant variables for each case of traced contacts and returning travellers. This figure now also illustrates graphically how we compute the fraction of transmission that is prevented by quarantine. We have also included a glossary table (Table 1 in Materials and methods) with a description of all variable names and their relationships.

Reviewer #3:Quantifying the impact of quarantine is an important topic, where the literature is currently lacking. This piece is very timely and tackles some very important questions in the space. At the moment, I feel this paper is somewhat lacking and requires some revision. I think the biggest issue is that it seems to be tailored for the situation in Switzerland. In itself, that's fine, but if the paper stays that way, the title should be altered to specify "in Switzerland", and the Introduction should give some more detail about the local situation. If the authors made that change, I have only a few technical queries that need to be addressed. I will also give some comments and suggestions about the differences between quarantine for close contacts compared to returned travellers, and what should be done to make the analysis more useful globally.

We thank the reviewer for their positive description of our project, and for the suggestions made. We feel the manuscript quality has greatly increased following these revisions.

With regards to the focus on Switzerland, the only country-specific statement that we make is that the baseline duration of quarantine to which we compare is 10 days. Other countries may use higher values, such as 14 or even 21 days, but under our framework we see that increasing quarantine beyond 10 days has very little (<1%) effect on transmission prevention.

For returning travellers, we are not considering the risk of being infected abroad versus at home. This risk should undoubtedly be included in an assessment of

whether to impose quarantine or not, as performed by Russell et al., 2020. But this is not the question we are asking. What we want to quantify is that given an individual is infected and that they have to go into quarantine, how do the different quarantine strategies perform? This question is independent of risk at home versus abroad. However, when it comes to choosing the optimal quarantine strategy, then the local epidemiological scenario will have an impact. We respond to this point in detail below.

1) I'm concerned that the calculations in this paper are incorrect. Specifically, I'm looking the second paragraph in subsection “Quantifying the benefit of quarantine” and Equation 1. If q(t) is the probability density of the generation time (and generation time is the time from exposure until the person becomes infectious), then the integral in Equation 1 is the probability that the person becomes infectious while in quarantine. I don't see how this is equal to the fraction of transmission prevented. I thought you would need the function to be the probability that someone is infectious at time t? Which I think is the convolution of Figure 2A and 2B. I am not too concerned about this, as if it is an error it shouldn't be hard to fix.

We apologise that the original text has caused some confusion between the generation time (the time interval between infection of an infector and the infection of a subsequent infectee) and the incubation period (the time interval between infection and symptom onset in the same individual). The generation time interval is exactly the probability density that an individual is infectious at time *t*, and this is what we use in our calculations of quarantine efficacy. Through clarifications to our notation, our new Figure 1, and the information content of the Materials and methods section, we hope that further confusion will be prevented.

In the Model description we now explicitly state: “In the model, the timing of onward transmission from an infected individual is determined by the generation time distribution, which describes the time interval between the infection of an infector and infectee (see Figure 1—figure supplement 1).”.

2) The other technical issue is about returned travellers, and how the trip duration affects the transmission reduction through quarantine. I'm not convinced by the argument that short trips need longer quarantine. I think the calculations are ok, and the result comes from the assumptions themselves. The authors are assuming that a traveller contracts covid while away, so on a 1 day trip, they must have caught it recently, and therefore need more time for symptoms to develop and to test positive. However, I don't think the authors should rule out a traveller catching covid before leaving, and becoming infectious after returning. I understand that the authors want to split the paper up into locally acquired cases, and international, but I don't think it works. If the situation was one where the country had no covid, then this claim about short trips needing longer quarantine makes sense. But in general it's a more nuanced question and certainly doesn't seem to fit Switzerland. I guess it could also work if you assume all local transmissions can be contact-traced, but I don't think the authors claimed this either.

We agree with the reviewer that the result about the relationship between the duration of the trip and the optimal duration of quarantine depends on the assumption that the infection is acquired during travel. This assumption is necessary because the concept of travel quarantines itself depends on the assumption. If local transmission rates are comparable to the travel destination, travel quarantines do not make sense. We now highlight in the Introduction that travel quarantine only applies to individuals who “have returned from recent travel to an area with community transmission that has been assessed to pose significant epidemiological risk (WHO, 2020).”, as specified in the WHO’s guidelines (WHO, 2020).

3) On to the more general point about applicability globally, which is specific to returning travellers (apart from the notes above, I think the close contact part of this work look good). I think the authors have missed explaining the decision context clearly (where the decision is about length of quarantine), and how differing values, objectives and system states affect things (see Baker et al., 2020 for a general discussion on decision making). The point I want to get to is, what is the objective of quarantine for returned travellers? The paper focusses on reduction of spread, which seems reasonable for countries with ongoing community transmission. However, it would be important to be clear about how much transmission in the country is coming through airports, relative to the community spread.There are many countries and jurisdictions that have either no ongoing transmission, or very well contained clusters. Quarantine in these areas is completely different, as the aim isn't about getting some percentage reduction is transmission. Instead, the aim is to have 100% reduction, and the metric of interest is the probability that an infection escapes quarantine and seeds a cluster of cases in the community. I think many people would be very interested in an analysis that looked at quarantine length and testing strategy in this context, to get towards a trade-off between length of stay, costs and importation risk.

What we think the reviewer is saying is that the optimal quarantine strategy depends on the goal you want to achieve, and in turn this goal depends on the current epidemiological scenario both at home and abroad.

In this work we have computed two measures of the usefulness of quarantine; firstly is the reduction of transmission, and secondly is this reduction divided by quarantine duration. In situations where we want to prevent the (re)introduction of SARS-CoV-2, we should focus on maximising the reduction of transmission (and hence minimising the transmission risk). If the disease is already endemic then we could be looking for trade-offs between transmission reduction and quarantine duration. Hence our two measures can be used to describe these scenarios. We have expanded on these points in the Discussion: “The ratio of transmission prevented versus days spent in quarantine is only one possible definition of utility. Defining the appropriate function is ultimately a policy question: the economic, societal, and individual costs are likely a function the days spent in quarantine, but we cannot determine the shape of this function. Furthermore, the local epidemiological situation will dictate which metric of quarantine efficacy is to be optimised. In situations where the goal is to prevent the (re)introduction of SARS-CoV-2, one should focus on maximising the reduction of transmission (and hence minimising the transmission risk). If the virus is already endemic then considering the trade-off between transmission reduction and quarantine duration could determine the optimum strategy. Another perspective is that the utility of preventing transmission is crucially dependent on whether it brings the effective reproductive number under one.”.

Overall, I think this paper has made some good steps. However, the message needs to be refined, and the context of the paper needs to be clarified. I think quarantine is a very important topic, and in its current state, the manuscript is only applicable to regions with active transmission. I think the current scope is acceptable at this journal, but it needs to be clear that it is aiming for that. Even without analysis suited to low/no prevalence scenarios, at a minimum there should be some discussion about how the local epidemiological situation is driving the results and the analysis, and how quarantine may need to be approached differently elsewhere.I wish the authors all the best in their revisions. As I said, it's an important topic and there needs to be more literature about it. I would happily review this paper again. I also commend the authors for making everything available on Github. This was useful during my review.

We thank the reviewer for their support, and we believe the message has been sharpened thanks to the combined comments of the Editor and reviewers. We are also happy to always share code and to be transparent about our methodology, sometimes even too transparent by including too many equations. The reviewer has made great points which have particularly improved the discussion and the overall message of this manuscript.

We have added the following paragraph to Results section where we begin to analyse returning travellers: “The rules for whether travellers returning from abroad are quarantined are frequently changed according to the epidemiological scenario in the travel destination and/or in the home country. A high risk of infection while abroad due to high prevalence, or the possibility of returning with a new virological variant, can lead to the imposition or reinstatement of quarantine measures. Countries that have already eliminated the infection may be even stricter in their quarantine approach to prevent new community-transmission clusters from being seeded. Here we do not discuss these scenarios or the concept of relative risk, we simply quantify how effective quarantine strategies would be at preventing transmission if the returning traveller was infected while abroad. Should quarantine rules be instated or modified, these results can help determine the optimal quarantine duration and/or testing strategy.”.

Furthermore, we have added the following paragraph to the Discussion to describe how the epidemiological scenario drives the choice of metric that defines the optimal quarantine strategy: “The ratio of transmission prevented versus days spent in quarantine is only one possible definition of utility. Defining the appropriate function is ultimately a policy question: the economic, societal, and individual costs are likely a function the days spent in quarantine, but we cannot determine the shape of this function. Furthermore, the local epidemiological situation will dictate which metric of quarantine efficacy is to be optimised. In situations where the goal is to prevent the (re)introduction of SARS-CoV-2, one should focus on maximising the reduction of transmission (and hence minimising the transmission risk). If the virus is already endemic then considering the trade-off between transmission reduction and quarantine duration could determine the optimum strategy. Another perspective is that the utility of preventing transmission is crucially dependent on whether it brings the effective reproductive number under one.”.

Reviewer #4:Ashcroft et al. discussed the appropriate duration of quarantine for travelers and close-contacts by estimating the fraction of transmission that can be avoided under a range of scenario and their relative utility. The study also explored the impact of testing, reinforced hygiene, adherence, and symptom presentation on transmission. This is an important and policy relevant topic. The article is nicely written, and I only have a few comments.Cases included in Kucirka et al. are all symptomatic, and they back calculated time from exposure assuming incubation period of 5 days. The authors here seem to assume that the sensitivity of the RT-PCR test found in symptomatic individuals can be applied to asymptomatic individuals. Please state this assumption (if that's correct) and discuss how this assumption impacts the outcome.

We thank the reviewer for clarifying how the sensitivity is calculated through the back calculation from symptom onset, as opposed to directly from the time of infection. The incubation period we use (from Ferretti et al., 2020) also has a median incubation period of 5 days, as used by Kucirka et al., 2020. We do assume the false-negative profile is the same across symptomatic and asymptomatic cases, which may not be the case. It is likely, although we couldn’t find any data on this, that the sensitivity is lower for asymptomatic cases, and hence we would be overestimating the efficacy of quarantine as the asymptomatic infections could be released earlier due to a more-probable false-negative test result.

This assumption is now stated when we describe the false negative rate in Materials and methods: “A further challenge with this false-negative rate is that it was calculated by Kucirka et al., 2020 from symptomatic cases only. Here we assume that this test sensitivity profile also applies to asymptomatic cases.”.

The impact of this assumption is discussed in the Results when we consider the role that symptomatic cases play in quarantine efficacy: “Furthermore, we have assumed that the false-negative rate is the same between symptomatic and asymptomatic cases. If the test is less sensitive (higher false-negative probability) for asymptomatic cases, then quarantine efficacy would be further reduced.”.

Finally, we acknowledge that this assumption is a limitation to our study in the Discussion: “Furthermore, these distributions, and also the test sensitivity profile, could be different between persistently asymptomatic and symptomatic individuals (Buitrago-Garcia et al., 2020), which ultimately lead to an overestimation of how much transmission is prevented by quarantine.”.

Figure 4 and other figures, can the authors state how the upper and lower bounds are estimated?

The bounding lines are now clearly described in all figure legends. By combining the standard quarantine with test-and-release in the results figures, we hope it is now clear that the standard quarantine protocol provides the lower bounds in Figures 2 and 4.